# A Hybrid-Model-Based CNC Machining Trajectory Error Prediction and Compensation Method

Wuwei He [1,2], Lipeng Zhang [1,2], Yi Hu [2,3], Zheng Zhou [1,2], Yusong Qiao [1,2] and Dong Yu [1,*]

1   University of Chinese Academy of Sciences, Beijing 100049, China; hewuwei@sict.ac.cn (W.H.);
    zhanglipeng@sict.ac.cn (L.Z.); zhouzheng18@mails.ucas.ac.cn (Z.Z.); qiaoyusong21@mails.ucas.ac.cn (Y.Q.)
2   Shenyang Institute of Computing Technology, Chinese Academy of Sciences, Shenyang 110168, China;
    huyi@sict.ac.cn
3   Shenyang CASNC Technology Co., Ltd., Shenyang 110168, China
*   Correspondence: yudong11@sict.ac.cn

**Abstract:** Intelligent manufacturing is the main direction of Industry 4.0, pointing towards the future development of manufacturing. The core component of intelligent manufacturing is the computer numerical control (CNC) system. Predicting and compensating for machining trajectory errors by controlling the CNC system's accuracy is of great significance in enhancing the efficiency, quality, and flexibility of intelligent manufacturing. Traditional machining trajectory error prediction and compensation methods make it challenging to consider the uncertainties that occur during the machining process, and they cannot meet the requirements of intelligent manufacturing with respect to the complexity and accuracy of process parameter optimization. In this paper, we propose a hybrid-model-based machining trajectory error prediction and compensation method to address these issues. Firstly, a digital twin framework for the CNC system, based on a hybrid model, was constructed. The machining trajectory error prediction and compensation mechanisms were then analyzed, and an artificial intelligence (AI) algorithm was used to predict the machining trajectory error. This error was then compensated for via the adaptive compensation method. Finally, the feasibility and effectiveness of the method were verified through specific experiments, and a realization case for this digital-twin-driven machining trajectory error prediction and compensation method was provided.

**Keywords:** intelligent manufacturing; digital twin; CNC system; trajectory error; artificial intelligence algorithm; prediction; compensation

## 1. Introduction

The global manufacturing industry is developing towards Industry 4.0 at a rapid pace [1]. Intelligent manufacturing is an important part of Industry 4.0, using advanced information technology and intelligent equipment to realize the intelligence and automation of the manufacturing process [2,3]. As one of the core elements of intelligent manufacturing, the CNC system has become an indispensable aspect of modern manufacturing, and its machining accuracy determines the level of industrial manufacturing achieved [4].

The machining trajectory error is the shortest distance from the current actual position of the tool to its desired trajectory and is mainly due to the inconsistency between the actual movement value of the tool relative to the workpiece and the command value; it can directly reflect the accuracy of CNC machining [5,6]. In the process of CNC machining, the influence of various factors, such as the accuracy of the machine tool itself [7], the following error generated by the servo system [8], the wear of the cutting tool [9], the deformation of the material [10], and so on, can lead to the generation of machining trajectory errors. In actual machining, these errors will directly affect the accuracy and quality of the machined parts, and even lead to machining failure. Therefore, machining accuracy is one of the critical indicators for evaluating the machining performance of CNC systems. The reasonable prediction and compensation of machining trajectory errors and control of machining

accuracy are of great significance for improving the machining performance of the CNC system and realizing high-speed and high-precision machining.

To improve the machining accuracy and enhance the machining performance of CNC systems, researchers have proposed a variety of CNC machining trajectory error prediction and compensation methods, which can be categorized into model-based and data-based methods [11,12]. A model-based method mainly describes the error characteristics by establishing a mathematical model, such as using polynomials, wavelet functions, etc., to represent the error change curve, and, through the optimal fitting of the model parameters, the machining trajectory error can be predicted and compensated for in the subsequent processing. The advantage of this method is that it can more accurately describe the change rule of the machining trajectory error. However, the model accuracy has a significant impact on the prediction and compensation results, especially since the mathematical model cannot accurately reflect the mapping relationship between the machining trajectory error and a variety of influencing factors such as motion control parameters, contour shapes, the CNC machining performance, etc. Establishing the model and optimizing its parameters requires a higher degree of sophistication [13].

Data-based methods utilize data analysis techniques for error prediction and compensation by collecting data from the actual machining process, such as machining process data, the cutting force, etc. Standard methods include regression analysis, neural networks, and so on. The advantage of these methods is that they can make full use of the information from the actual data and have better adaptability to complex error characteristics. However, on the one hand, a large amount of experimental data and robust data processing capabilities are required. On the other hand, these methods are affected by the sampling frequency of the data. Problems due to the inaccuracy of the prediction and compensation will occur at low sampling frequencies [14].

Digital twin technology is one of the core technologies of cyber–physical systems (CPSs) and one of the critical enabling technologies for intelligent manufacturing; it has gradually received extensive attention from academia and the industry [15–17]. The bidirectional data flow between the virtual and physical spaces in digital twin technology ensures state updates in the virtual space and control in the physical space. For example, the prediction and optimization of the machining state, machining results, etc., are achieved through data analysis across sensors and equipment during the operation of the CNC system [18]. Tong et al. [19] proposed a real-time machining data application service for intelligent CNC machine tools, including multi-sensor fusion technology, an MTConnect protocol, and a developed human–machine interface. Data analysis and the optimization of the machining processes, such as the machining status, machining trajectory, and energy consumption, were achieved by building a digital twin model. Zhao et al. [20] proposed a cutting parameter optimization method by constructing a digital twin model of a CNC machining tool. The model utilizes the simulation and optimization of the virtual twin, combined with real-time sensing and dynamic optimization of the machining process parameters, to achieve a reduction of carbon emissions during machining. However, in actual production, how to effectively integrate monitoring, prediction, and optimization functions based on digital twin technology to achieve the prediction and compensation of machining trajectory errors driven by the model and data together still requires further research and exploration.

Based on the above discussions, in this study, we transformed the CNC twin into an independent unit with automation, bi-directional command transmission, and a feedback information data flow by constructing a digital twin model of the CNC based on a hybrid model. Through this approach, model-based and data-based methods can be effectively combined to solve the problems of insufficient accuracy and low efficiency in the prediction and compensation of machining trajectory errors. Therefore, digital twin technology provides a new method for realizing CNC machining trajectory error predictions and compensations. The main contributions of this study are as follows:

- A hybrid-model-based digital twin framework is proposed for CNC systems.

- A neural-network-model-based machining-trajectory-error-tracking prediction algorithm is proposed.
- An adaptive error compensation method is proposed for the machining trajectory error according to the tracking error prediction results.
- An application example of the machining trajectory error prediction and compensation method is given under the CNC system's hybrid-model-based digital twin framework to verify its feasibility and superiority.

The rest of this article is organized as follows: Section 2 contains the problem description and Section 3 presents a hybrid-model-based digital twin framework for CNC systems. Machining trajectory error prediction and compensation methods are investigated in Section 4. Section 5 demonstrates a specific system's implementation and experimental validation. The conclusion of this paper is described in Section 6. Finally, Section 7 contains a relevant discussion of this paper and current research.

## 2. Related Work

### 2.1. The Changes of CNC Systems in Intelligent Manufacturing

Intelligent manufacturing refers to a new type of manufacturing that can create and deliver products and services through the integrated and intelligent utilization of processes and resources in the information and physical spaces of different system hierarchies [21,22]. With the promotion and application of intelligent manufacturing technology in aerospace and other precision machining fields, intelligent application service demands such as self-awareness, self-decision-making, self-execution, self-learning, and self-optimization have been put made of CNC systems [23]. To cope with the new demand for intelligent manufacturing, CNC systems are not only gradually transitioning from their traditional closed structure in the direction of open, digital, and intelligent systems, but they are also changing in terms of their characteristics and structure.

In recent years, the proposed standard GB/T 40647-2021 [24] "Intelligent Manufacturing System Architecture" has stipulated three dimensions of an intelligent manufacturing system's architecture: life cycle, system level, and intelligent features [25]. A CNC system and its system architecture, with respect to the specific changes occurring, are shown in Figure 1.

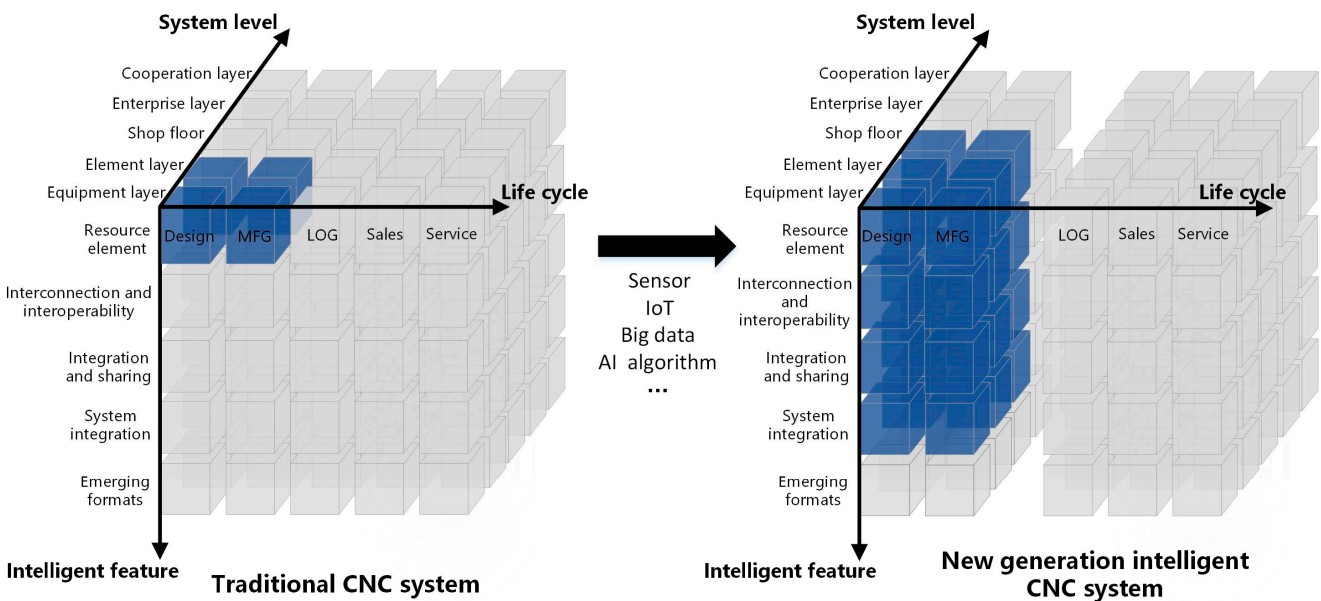

**Figure 1.** The changes to CNC systems in intelligent manufacturing [25].

In the intelligent manufacturing system's architecture, the system level dimension is its level, from on-site equipment to collaborative manufacturing; its life cycle dimension is

the life cycle of its products, equipment, and services; and its intelligent feature dimension reflects the degree of intelligence of the manufacturing process, which represents the objective law and figurative expression of the intelligent manufacturing. The traditional CNC system, in the system hierarchy dimension, is the equipment and unit layers and only contains resource elements in the intelligent feature dimension. The new generation of intelligent CNC systems expands to the shop floor in the system dimension. It has the characteristics of interconnection, integration, sharing, and system integration as its intelligent features.

However, conventional CNC systems rely solely on controllers, drives, and machine tools to carry out machining tasks. Due to the complexity of the process and the challenge of fully utilizing the available data, it can often be difficult to accurately analyze the correlation between the issues present in the machining results and the corresponding machining steps [26].

The new generation of CNC systems plays an essential role in digital workshops and intelligent factories as a flexible production unit [27]. Hence, functional applications such as site sensing, data processing, and value assessment should be equipped with a CNC system in intelligent manufacturing. These intelligent functions can improve machining performance, making it easier to solve the problem of measuring and compensating for machining trajectory errors during the machining process due to changes in time and working conditions.

In this paper, based on the structure of the current CNC systems, digital twin technology is introduced to sense and analyze the data in the machining process, which links the functions of sensing, prediction, and analysis to the existing CNC system. In order to cater to the requirements of intelligent manufacturing, it is important to establish a smart CNC system hierarchy, as depicted in Figure 2. This not only fulfills the new demands of intelligent manufacturing but also supports the enhancement of machining precision.

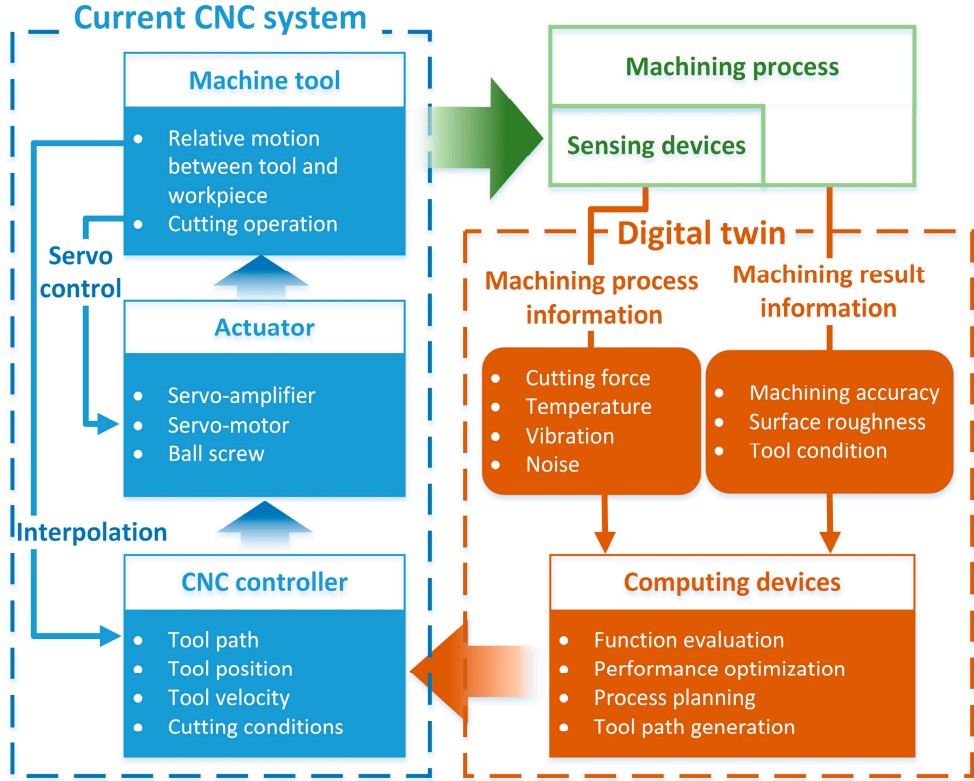

**Figure 2.** Hierarchical structure of intelligent CNC systems.

## 2.2. Machining Trajectory Error Prediction and Compensation

CNC machining is programmed using G-code, which consists of non-cutting operations such as representing basic geometric movements, axis address spaces, tool changes, or coolant switches. The G-code is interpreted and executed by the CNC system, which controls the movement of the tool along a straight line, arc, or helix on the machined part to form the machining trajectory. However, as the following error of each axis of motion is affected by factors such as the mechanical structure and machining parameters, the errors with respect to each axis are not the same; the actual machining trajectory has an error compared to the theoretical machining trajectory, which seriously affects machining accuracy.

Scholars have conducted many studies to improve the accuracy of machining trajectory error prediction and compensation. Chen et al. [28] used the Frenet framework with linear computational complexity and proposed an analytical processing method for the linearization of trajectory error with speed constraints based on the upper limit of the feed rate computed by the framework. Huo et al. [29] used a pre-trained nonlinear autoregressive network with external inputs (NARX) to predict the machining trajectory of a CNC system. Based on their prediction results, they determined the compensation term that needed to be added at the reference input position to reduce the trajectory error and used simulation experiments with linear, circular, and parabolic contours to validate its effectiveness in trajectory error prediction and compensation. Li et al. [30] combined neural network modeling to propose a long short-term memory neural network (LSTM-NN)-based machining trajectory error estimation and compensation method, which utilizes the LSTM-NN model to predict the tracking error, combines contour detection algorithms and sensor data, and monitors the morphological changes on the surface of the workpiece in real time to estimate the contour error and compensate for it. The realization of these works can improve the ability of a CNC system to deal with trajectory errors, but there are still problems, such as low efficiency and a lack of accuracy.

## 2.3. Research Motivation

According to a recent study, researchers are exploring advanced technologies, such as digital twins, artificial intelligence, and new information and communication technologies, to conduct intelligent research on CNC systems in intelligent manufacturing. The aim is to improve the machining performance of CNC systems. However, the current research has some areas for improvement:

1.  Currently, research on digital twin modeling, information perception, and fusion for CNC systems primarily involves conceptual, architectural, or qualitative analyses, which lack specific theoretical methods and critical technology research results.
2.  A CNC system is a kind of mechatronic equipment, using multiple devices to complete a whole set of machining processes in a unified and coordinated manner. The traditional modeling method is often only used for a part of the CNC system, for individual modeling, and seldom considers the kinematic chain, servo dynamics, and other related information in the modeling process.
3.  To achieve high-speed and high-precision machining, existing methods to process machining trajectory errors require improved processing speed and accuracy.

In this paper, we propose a solution to the drawbacks faced by CNC systems by constructing a hybrid-model-based digital twin framework. The framework includes multiple models based on the CNC system and its auxiliary system in the physical space and uses the digital twin platform in the information space as its core. The hybrid model uses various models, such as kinematic chain and dynamics models, combined with AI algorithms to enable data storage, data preprocessing, performance analysis, optimization decision-making, knowledge learning, and dynamic execution in the machining process. By improving the ability of CNC machining trajectory error prediction and compensation, this model supports the machining performance improvement of CNC systems and the realization of intelligent manufacturing.

## 3. A Hybrid-Model-Based Digital Twin Framework for CNC Systems

### 3.1. General Structure of the Framework

A CNC system is a type of mechatronic equipment comprising three main components: the machine tool, the servo system, and the CNC device [31]. The machine tool is the mechanical execution part of the CNC system, comprising the machine bed, table, tool holder, spindle, chuck, tailstock, and other parts. The servo system includes motors, drive control systems, and other components that drive the machine tool. The CNC device is the control system responsible for controlling the processing and operation of the machine tool, which is the key to achieving automated processing.

In this paper, we have summarized the above elements into three parts: mechanical, electromechanical, and control systems. The mechanism is as follows: the mechanical system unit consists of a servo motor that makes the moving parts move through power transmission. The electromechanical system unit drives the servo motor through speed control and torque control and converts the mechanical movement of the machine elements into electrical signals, which are fed back to the control system for processing. The motion control function of the control system unit sends commands, which are converted into electrical signals sent to the motor drive system, while the servo's feedback is analyzed. Different parts of the CNC system have their own functions. Three units describe the interaction between the components of the CNC system, as shown in Figure 3.

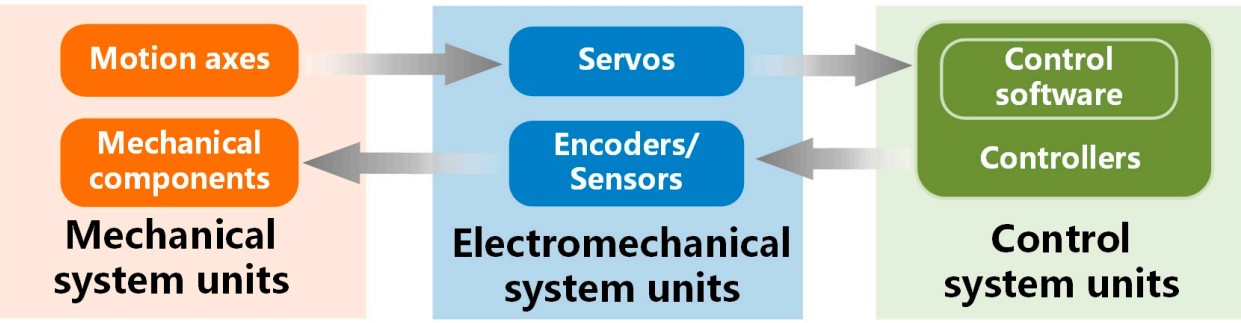

**Figure 3.** The interrelationships between the components of CNC systems.

Combined with the structural relationship between the various parts of the CNC system, we propose a hybrid-model-based digital twin framework for the CNC system. Its specific framework structure is shown in Figure 4, which shows a hybrid model integrating a kinematic chain model and dynamics model for the three units of the CNC system mentioned above, and it also includes an AI algorithm, virtual machine tool, and visualization client to realize processing with respect to data sensing, evaluation, optimization decision-making, and data visualization. The red arrow in the figure represents data transmission, and the green arrow represents result feedback. The model solves the problems in generating, executing, and analyzing the machining tasks of the CNC system using three subsystems. In addition, all control and analysis tasks are performed on the digital twin model of the CNC, which allows for the enhanced sensing, analyzing, evaluating, and decision making of the CNC machining process.

The digital twin framework of CNC systems based on a hybrid model mainly consists of two parts: the physical space and the cyber space. The physical space and cyber space interact and integrate virtual and real data through digital threads (such as OPC UA and MTConnect) and IoT technologies, based on different task requirements. The specific details are described as follows:

Physical space: This primarily includes the physical entities of the control system, electromechanical system, and mechanical system. In this space, the control system sends control signals to the electromechanical system, receives feedback from it, and the electromechanical system outputs torque information to the mechanical system while responding to changes in load such as working load and the moment of inertia.

Cyber space: This space mainly comprises five components: a kinematic chain model, dynamic model, digital twin database, AI algorithms, and a virtual machine tool. The kinematic chain model and dynamic model represent the mechanical system and electromechanical system virtually through a multidisciplinary unified modeling approach. Detailed descriptions of these models will be provided in Sections 3.3 and 3.4 Some typical intelligent functions of CNC systems, such as machining trajectory error compensation and prediction, virtual debugging, and fault diagnosis, require data analysis and storage. Therefore, a digital twin database is used to store the information generated during the machining process in both the physical and virtual spaces, and AI algorithms are utilized for data analysis. The virtual machine tool includes the digital twin model of the machine tool and a simulation of the machining process control system. These components simulate the actual machining process in the cyber space, and, combined with virtual mapping strategies, they optimize and debug the machining process parameters in the physical space based on simulation results, achieving performance optimization in the complex machining scenarios of CNC systems.

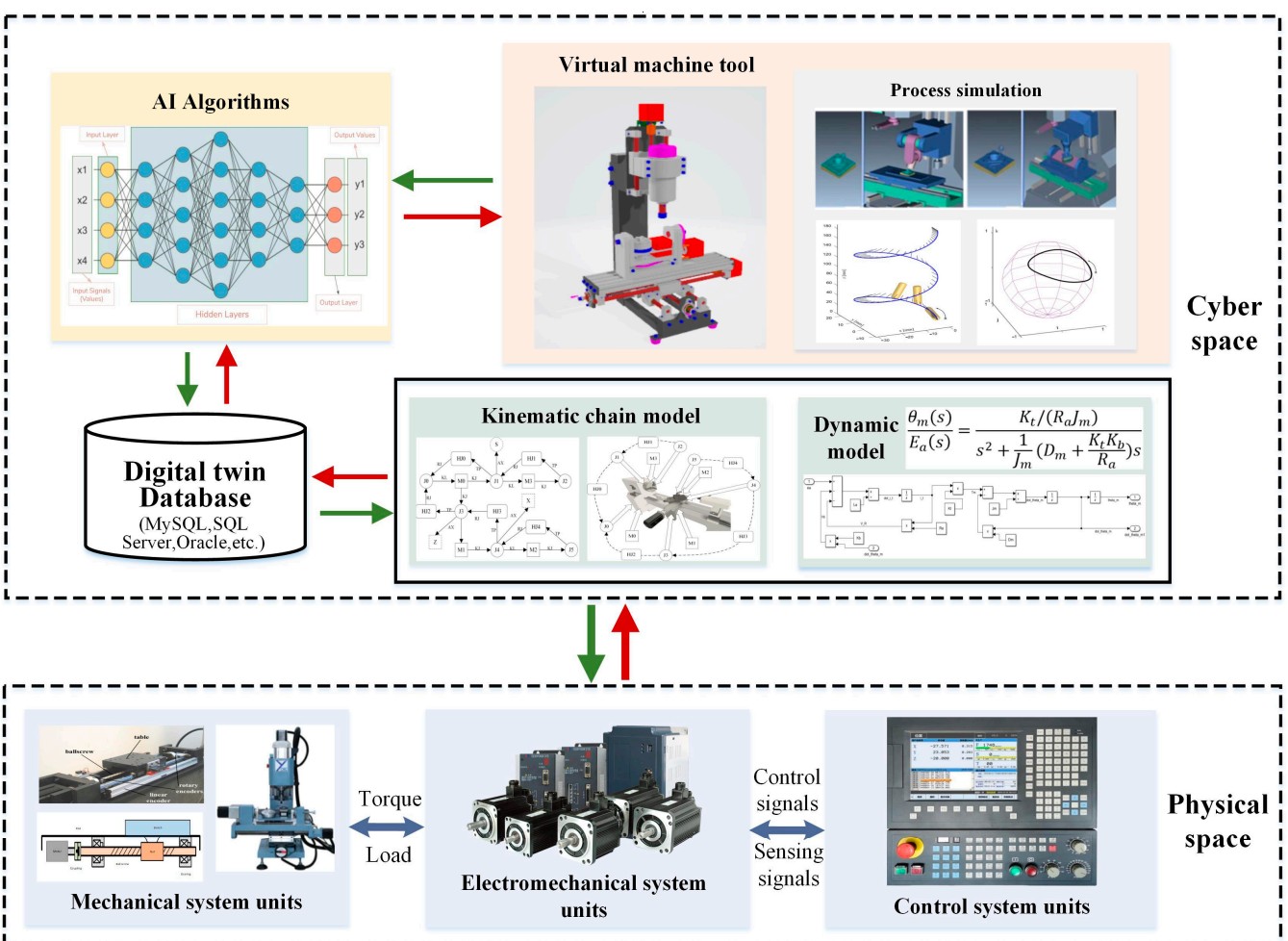

**Figure 4.** A hybrid-model-based digital twin framework for CNC systems.

As a typical mechatronics product, the CNC system involves various disciplines such as mechanical engineering, electrical engineering, and control system engineering. It exhibits characteristics such as multivariability, nonlinearity, and strong coupling, making the modeling and simulation of CNC systems particularly challenging. Traditional single-domain simulation tools are insufficient to meet the requirements for analyzing the overall performance of complex systems. Therefore, it is necessary to employ multi-domain modeling and simulation technology to complete the model development process. Additionally,

we enhanced the analysis and decision-making capabilities of the models through digital twin technology and artificial intelligence. The steps for model development are as follows:

1.  Decomposition of the overall system: Considering the relationships among various components of the CNC system in the physical space, the system is decomposed into three subsystem models—mechanical, electrical, and control. Mechanistic analysis is performed on these subsystems.
2.  Construction of the information space: Based on the mechanisms of the subsystems in the physical space, the role of the information space in the intelligent functions of the CNC system is analyzed. Using the Modelica multi-domain unified modeling language, the operational mechanisms of each subsystem are compiled and described to establish a digital twin database. This model is then updated and optimized through AI algorithms and the machining simulation of the virtual machine tool, achieving a virtual-to-real mapping between the physical and information spaces. This process results in the creation of a digital twin model of the CNC system, ensuring good consistency between the physical operation and model response.
3.  Communication between subsystems: The coupling relationships between the subsystems are analyzed, and their coupling mechanisms are studied to construct coupling interfaces between the subsystems. Digital threads, IoT, and other technologies are utilized to realize the coupling connections between the subsystems, thereby establishing the digital twin framework of the CNC system based on its hybrid model.

The framework is a digital representation of CNC systems in the physical world, aiming to describe the nature of the real world accurately. It enables an integrated solution for designing, processing, controlling, analyzing, and optimizing CNC machining. It will provide critical support for CNC research and applications and play a key role in practice. The framework can provide the following advantages for machining trajectory error prediction and compensation:

*   The improvement of workpiece machining quality and time by simulating the machining process and optimizing the process parameters;
*   Reducing the time and cost of machining trajectory error prediction and compensation, thus scaling up machining;
*   Increasing the efficiency of identifying the source of problems when issues arise with the processes and equipment.

### 3.2. Framework Modeling Approach

In order to satisfy the high fidelity and consistency of the model, a specific model construction principle needs to be adopted to realize the construction of the digital twin framework of the CNC system. Although the traditional data-flow-based modeling method can reveal the physical laws of the system, it is challenging combine the actual interaction characteristics between multiple models of the CNC system in the framework. When the original system structure changes or is replaced, the model requires modifications.

The digital twin modeling process for CNC systems involves abstracting, simplifying, describing, processing, and storing real-world physical entities. This object-oriented method is a way of modeling a problem domain with an object-oriented view, in accordance with how people are used to thinking. Its fundamental goal is to narrow the "semantic gap" between the system and the natural world using the same terminology as the performed functions.

The construction method of the framework is shown in Figure 5. First, the CNC system is decomposed into several classes, including its mechanical system, electromechanical system, and control system, using object-oriented modeling [32]. Then, the mathematical descriptions of the attributes and methods of the models of each system type, the number of sub-modules, and finally the attributes and methods of each sub-system and sub-module are determined. The numbers 1 and * in the figure indicate the relationship between classes, for example, an axis class can contain multiple linear axis classes or rotation axis classes.

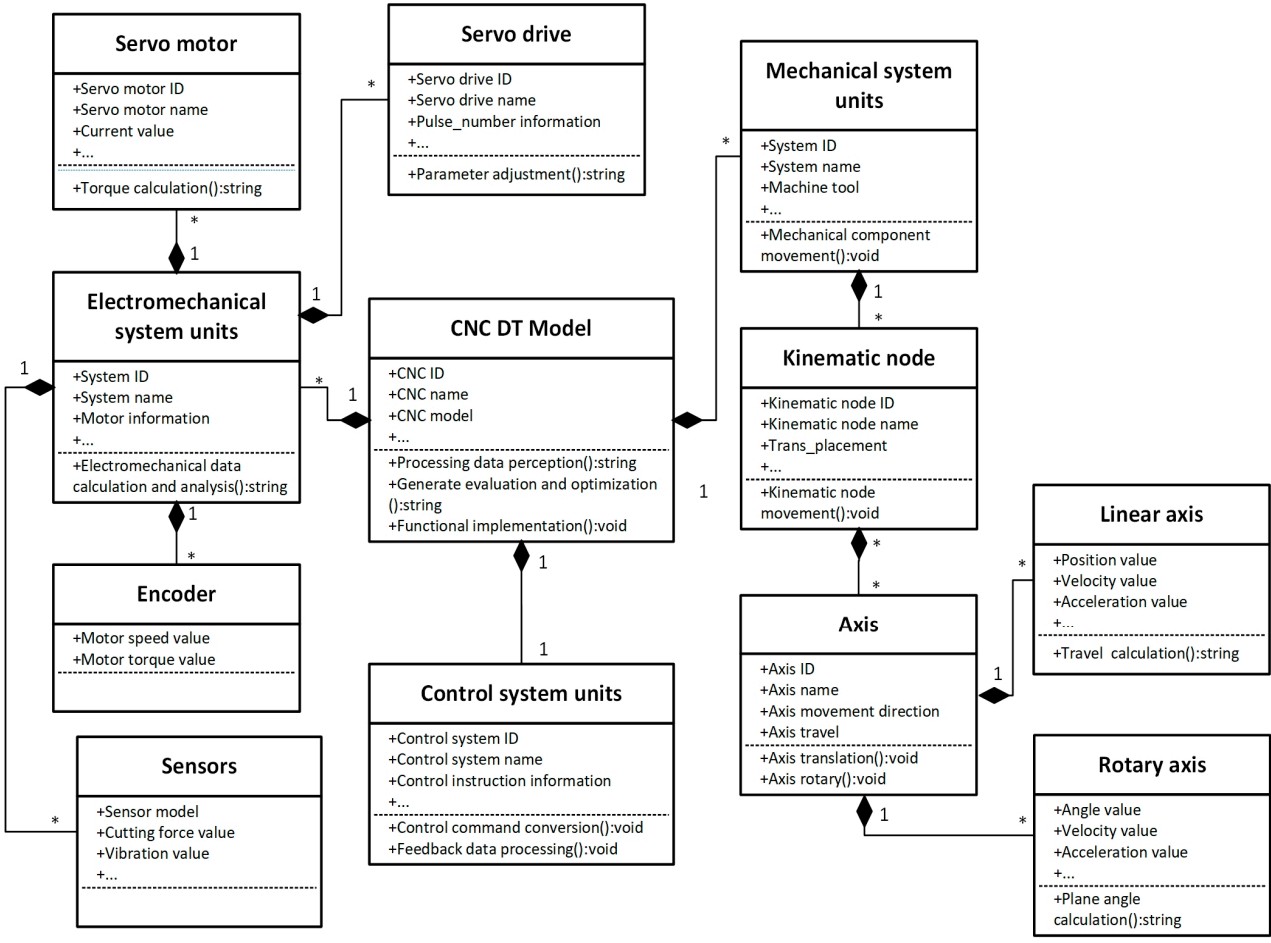

**Figure 5.** The modeling method for hybrid-model-based digital twins of CNC systems.

*3.3. Kinematic Chain Model*

In the digital twin framework, the mechanical system unit is a kinematic chain consisting of mechanical components, kinematic axes, and the workpieces between them. The mechanical components include the body, slide, table, and guideway of the machine tool, and the kinematic axes mainly include feed axes and rotary axes, which can be regarded as independent kinematic nodes of the above elements. In order to accurately describe the motion relationship between the motion nodes in the unit in the virtual model, a kinematic chain model can be established.

In the modeling process, since the mechanical system unit contains the parallel motions of many kinematic nodes, it is necessary to refer to the parent nodes and their forward kinematic chains. However, traditional kinematic chain modeling methods such as D-H representation and directed graphs cannot describe the parallel kinematic chains between kinematic nodes well. Therefore, we use kinematic chain representation using the recursive backlinking method (RBM) to solve the above problem. This method adequately represents the parallel kinematic relationships between the components in a mechanical system unit by storing a backward reference to the parent node in each kinematic node. The schematic diagram of the method is shown in Figure 6, where green arrows represent the links to child nodes and red arrows represent the references to parent nodes.

In the machining process, most of the movements of the mechanical system units are realized by serial movements by expressing the serial kinematics chain as equations; the position of the axes are the variables, the position and orientation of the tool concerning the workpiece are the parameters, and the actual machining information is the result. In the digital twin modeling process, if acceptable values can be found for all the variables while satisfying these conditions, then the mechanical system unit in the model is in a valid state.

Therefore, establishing a complete and adequate mechanical system unit model is crucial for motion analysis, trajectory planning, and control realization of the CNC machining process.

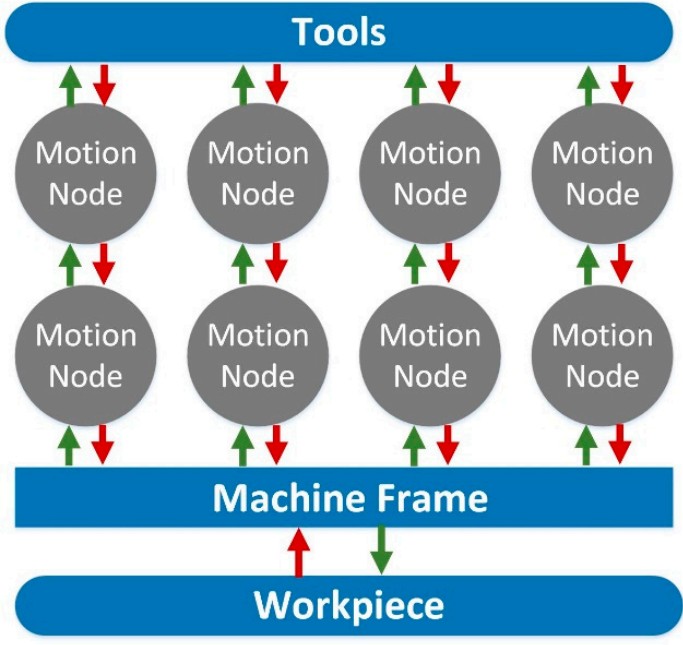

**Figure 6.** RMB-based kinematic chain modeling.

### 3.4. Dynamics Model

The electromechanical system unit is a complex system of electromechanical coupling, and the study of its dynamics model can provide feedback on the data contained in the machining process of each axis of motion. Taking the servo motor controlled by the armature in the CNC system as an example, the mechanism model is shown in Figure 7. The simulation and analysis of the CNC machining process through a digital twin framework containing a dynamics model is of great significance for compensating CNC machining trajectory error prediction.

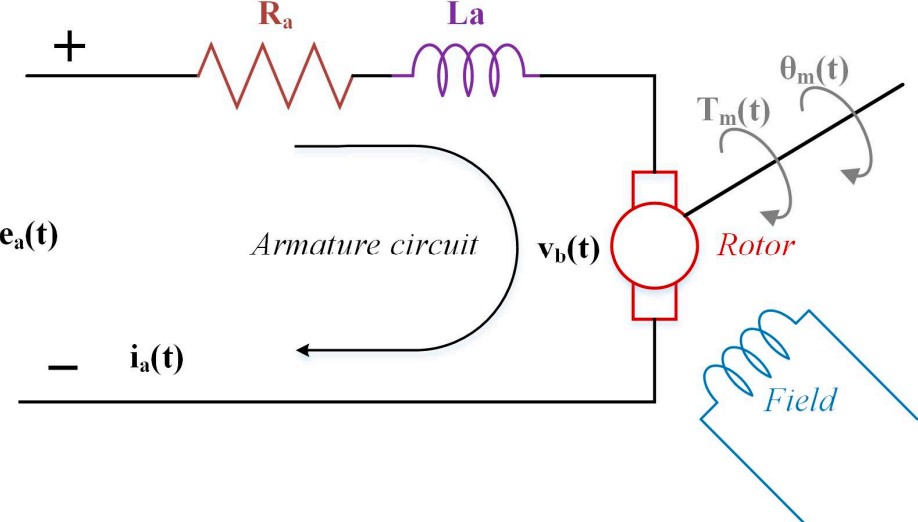

**Figure 7.** Electromechanical system unit's mechanism model.

Where $i_a(t)$, $e_a(t)$, and $v_b(t)$ denote the armature current, voltage, and reaction potential, respectively, the equilibrium equation of the armature circuit in the motor is expressed by the Laplace transform:

$$E_a(s) = R_a I_a(s) + L_a s I_a(s) + V_b(s) \tag{1}$$

The relationship between the reverse electromotive force and the angular velocity of the motor is:

$$V_b(s) = K_b \frac{d\theta_m(s)}{dt} \tag{2}$$

where $K_b$ is the proportionality constant, usually called the reverse electromotive force constant, and $\theta_m(s)$ is the angular velocity of the motor, which can be obtained by performing the Laplace transform of Equation (2):

$$V_b(s) = K_b s \theta_m(s) \tag{3}$$

Meanwhile, the torque produced by the motor as a function of the armature current can be expressed as follows:

$$T_m(s) = K_t I_a(s) \tag{4}$$

where $K_t$ is the constant of proportionality, usually called the motor torque constant; the value of the parameter is determined by the motor and magnetic field characteristics; and $T_m(s)$ is the motor torque. By substituting Equations (3) and (4) into Equation (1), the kinetic model transfer function can be obtained as follows:

$$E_a(s) = \frac{(R_a + L_a s)T_m(s)}{K_t} + K_b s \theta_m(s) \tag{5}$$

Also, considering the equivalent mechanical load in the electromechanical system, we can obtain

$$T_m(s) = (J_m s^2 + B_m s)\theta_m(s) \tag{6}$$

where $J_m$ and $B_m$ are the total rotational inertia and system damping terms. Substituting Equation (6) into Equation (5) yields

$$E_a(s) = \frac{(R_a + L_a s)(J_m s^2 + B_m s)\theta_m(s)}{K_t} + K_b s \theta_m(s) \tag{7}$$

The simplification of Equation (7) can establish the electromechanical system dynamics model transfer function as follows:

$$\frac{\theta_m(s)}{E_a(s)} = \frac{K_t / (R_a J_m)}{s^2 + 1/J_m(D_m + K_t K_b / R_a)s} \tag{8}$$

The simulation model of the electromechanical system dynamics is shown in Figure 8.

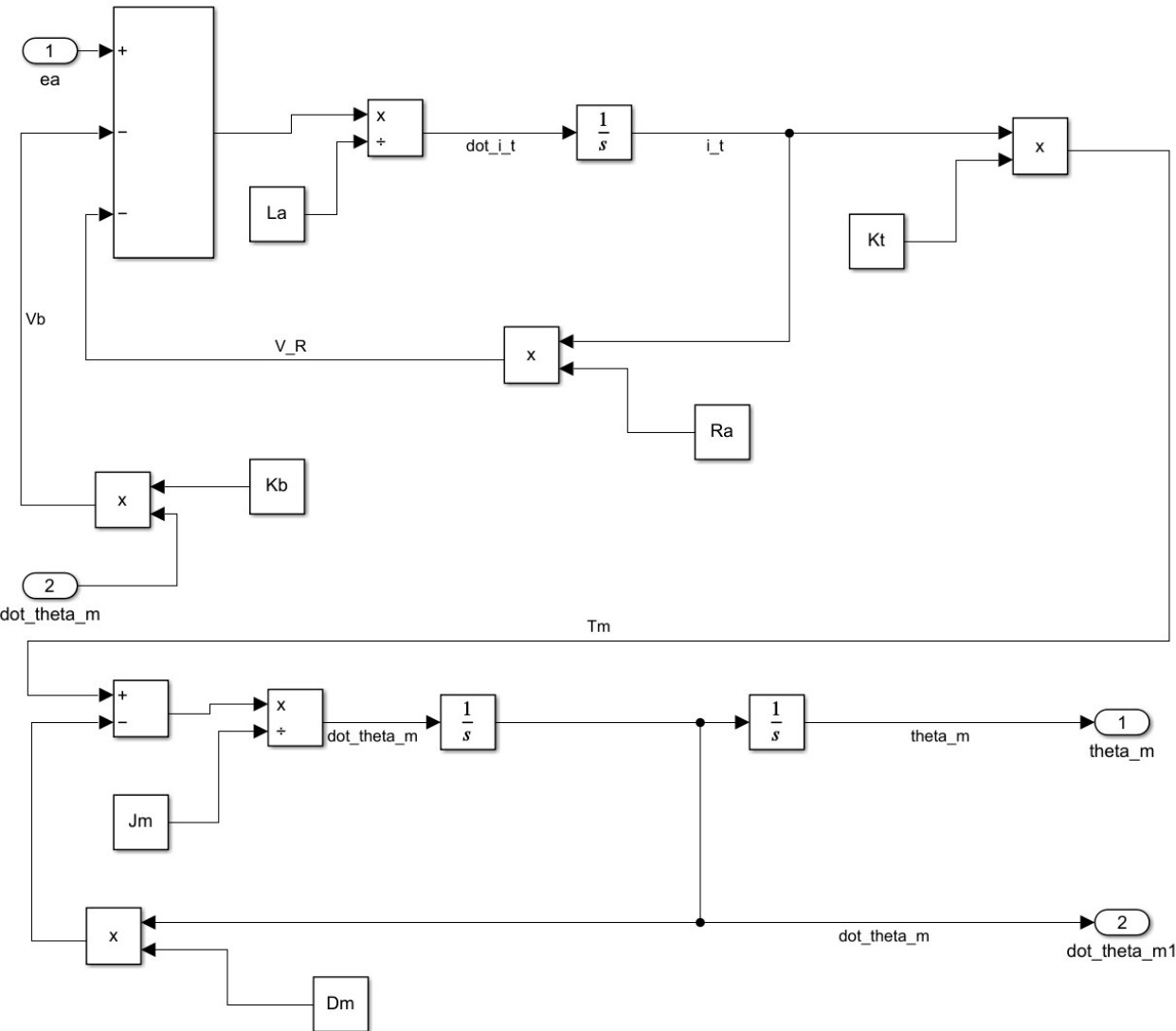

**Figure 8.** Electromechanical system dynamics simulation model.

## 4. Machining Trajectory Error Prediction and Compensation Method

### 4.1. Mechanism Analysis and Solution

As an essential performance indicator of the CNC system, machining accuracy is not only affected by the conversion of program segments but is also constrained by various mechanical and electrical factors. Hence, the existence of machining trajectory errors is inevitable. Machining trajectory errors can be measured using measuring instruments (such as coordinate measuring machines, scales, laser scanners, etc.) during the actual measurement of the processed parts. However, considerable manual effort is required to use sophisticated measuring equipment and methods to obtain this information.

The tracking error is the position deviation between the actual position point of the feed axis and the corresponding command position point, which can be expressed as the difference between its theoretical position and actual position in the CNC machining process and can be obtained using the data acquisition device on the field bus [33]. The machining trajectory error is the shortest distance between the actual position and the trajectory of the tool. The machining trajectory error is both related to and different from the tracking error, and their relationship is shown in Figure 9.

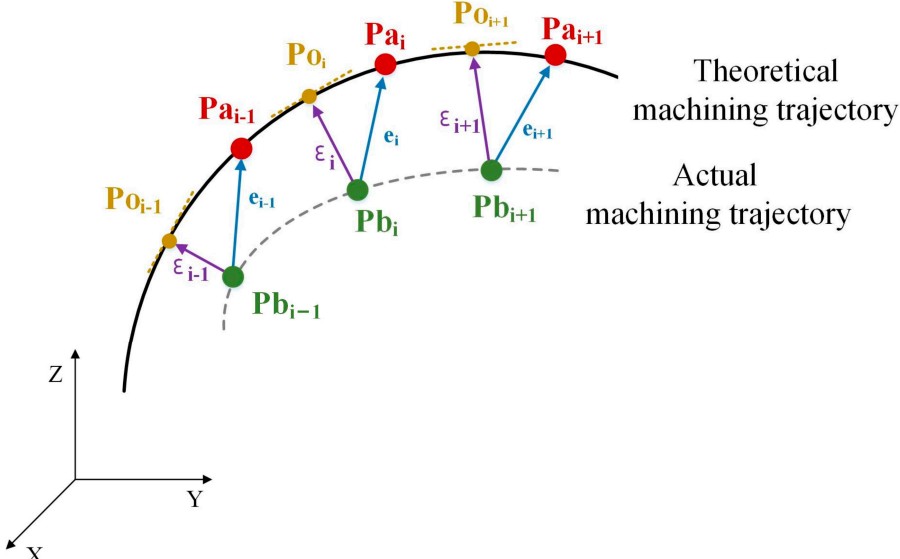

**Figure 9.** Machining trajectory error formation mechanism.

Where the points $Pa_{i-1}$, $Pa_i$, and $Pa_{i+1}(i = 1, 2, 3, \ldots)$ are the theoretical command position points and the actual position points are $Pb_{i-1}$, $Pb_{i-1}$, and $Pb_{i+1}(i = 1, 2, 3, \ldots)$, the tracking errors $e_{i-1}$, $e_i$, and $e_{i+1}$ in the direction of the x, y, and z axes exist in the spatial coordinate system and can be obtained:

$$e_i^\kappa = Pa_i^\kappa - Pb_i^\kappa, \kappa = x, y, z \tag{9}$$

where $Po_{i-1}$, $Po_i$, and $Po_{i+1}$ are the nearest points of the actual position point to the theoretical machining trajectory, which is usually called the plumbing point, the machining trajectory errors of $\varepsilon_{i-1}$, $\varepsilon_i$, and $\varepsilon_{i+1}$ can be expressed as follows:

$$\varepsilon_i^\kappa = || \overrightarrow{Pb_i^\kappa Po_i^\kappa} ||, \kappa = x, y, z \tag{10}$$

As shown in Figure 9, if the value of the tracking error is minimal and can be ignored, the value of the machining trajectory error must be minimal. However, if the value of the machining trajectory error is approximately zero, the tracking error is not necessarily zero. Therefore, in the actual machining process for a motion system, the control goal is to minimize the machining trajectory error to improve machining accuracy. In this paper, we use an AI algorithm to calculate the specific steps of the machining trajectory prediction and compensation method as follows:

1.  Given the machining trajectory data and data acquisition time interval for each axis, perform data preprocessing and feature extraction (described in detail in Sections 4.2 and 4.3);
2.  Predict the tracking error based on the proposed AI algorithm and obtain the tracking error value $e_i^\kappa$ for each axis (described in detail in Section 4.4);
3.  Calculate the actual position points based on the theoretical machining trajectories and the predicted obtained tracking error for each axis, and calculate the actual position point coordinate equations as follows:

$$Pb_i^\kappa = Pa_i^\kappa - e_i^\kappa, \kappa = x, y, z \tag{11}$$

4.  Perform machining trajectory position estimation from the predicted actual and theoretical positions. The actual position is compensated according to the adaptive dynamic error compensation method (described in detail in Section 4.5);

5. Output the compensated G-code to verify the effectiveness of machining trajectory error prediction and compensation.

### *4.2. Data Processing*

Due to the variety and complex composition of the machining trajectory data generated during machining, there are problems such as missing data and noise. In addition, the magnitude of these data usually varies, and the features associated with trajectory errors often account for a small proportion of the raw data due to their small magnitude. Therefore, it may be challenging to analyze the raw data directly to find the correlations between the data accurately [34].

After the above analysis, the raw processing trajectory data need to be preprocessed to remove the data that are not useful for error prediction compensation, and data noise reduction is then performed to eliminate various types of interferences effectively. To accurately predict and compensate for errors, the data need to be standardized to eliminate any scale-related influences. This highlights the significance of trajectory-error-related features and allows for a more precise correlation between the data.

### 4.2.1. Missing Data Processing

In the process of machining trajectory error prediction and compensation, data acquisition is a critical step. However, in practice, missing data issues often occur in the acquisition process for various reasons. Missing machining trajectory data refer to the existence of random missing values in the collected data, i.e., the values of some data points or periods are not recorded or not acquired.

Missing data pose challenges for the subsequent analysis and modeling of processing trajectories. First, the existence of missing values leads to the incompleteness of the dataset, which may affect the data's overall distribution and statistical characteristics. Second, to ensure the accuracy and reliability of the subsequent analysis, the processing of the missing values requires selecting appropriate methods for their filling or estimation.

Since machining trajectory data are time-series-related, the values in their features vary over their acquisition time and are strongly correlated. Compared with the direct use of the mean and median, the kNNI (k-nearest neighbors imputer) preprocessing technique identifies neighboring points via distance measurements. It can use the complete values of adjacent observations to estimate missing values [35]. Therefore, in this study, the missing data are processed using the kNNI missing value processing method, which can be expressed as follows:

$$C_x = \frac{\sum_{i=1}^{k} C_i}{k} \tag{12}$$

where $C_x$ is the missing value, and $C_i$ is the eigenvalue near the missing value. The method searches for k nearest neighbor samples through the Euclidean distance matrix. It fills the missing values using the mean of the non-null values of the corresponding positions of the closest neighboring samples.

### 4.2.2. Data Noise Reduction

Smoothness and stability are the desired states in actual machining. The machining motion trajectory is designed to be as smooth as possible. The speed and acceleration of the feed system during machining are required to have a smooth transition to reduce fluctuations and jumps. However, in the collected machining data (such as actual speed, actual position, following the error, etc.), some high-frequency fluctuation values are characterized by their small fluctuation amplitude, high frequency, and small regularity. High-frequency fluctuations may cause disturbances in subsequent data analysis, ultimately affecting machining accuracy.

Moving average (MA) is a signal smoothing method in the time domain that can effectively remove high-frequency fluctuations. In this paper, the MA is used to reduce noise by taking a data volume of length 3 for averaging, which can be expressed as follows:

$$F(n+1) = F(n) - \frac{f(n-1)}{3} + \frac{f(n+2)}{3} \tag{13}$$

where F(n+1) is the result after noise reduction and f(n) is the original data sequence. This method can effectively achieve data noise reduction by removing the high-frequency fluctuations and obtaining low-frequency data for model training.

### 4.2.3. Dimensionless Data

Machining trajectory data contains the command position, command speed, and command acceleration generated by the CNC system, and the actual position, actual speed, actual acceleration, and tracking error fed back by the servo drive. These data vary in order of magnitude in terms of range and units. A direct analysis is more likely to overlook some of the smaller order of magnitude indicator data, affecting the results of the data analysis.

To eliminate the influence of different features, this paper normalizes trajectory data processing. This normalization can accelerate the gradient descent speed in the model training process and improve the convergence of the model with the following formula:

$$x^* = \frac{x_i - x_{min}}{x_{max} - x_{min}} \tag{14}$$

where $x_i$ is the machining trajectory data value, $x^*$ is the normalized value, and $x_{max}$ and $x_{min}$ are the maximum and minimum values of the data series before normalization.

### 4.3. Feature Extraction

The operating data records generated by the CNC system during the machining process have important characteristic information, which reflects the corresponding mapping relationship between the input and output of the system. By analyzing and processing the machining trajectory data, we can reveal the causal relationship between the data and the effect of different features on the system's response. This is important for improving the accuracy of machining trajectory error prediction and compensation.

In the actual machining environment, analyzing the features of the trajectory trend will help accurately predict the machining trajectory error. Traditional feature extraction mainly includes position information with respect to each axis, curvature, speed, acceleration. However, the monotonicity and trend of these statistical features are often unsatisfactory compared to the rapid trajectory change trend and need to characterize the characteristic trend of trajectory change better.

To address this problem, we adopt a cumulative feature transformation method, i.e., transforming the extracted features into their corresponding cumulative form. Specifically, the incremental transform is carried out by applying a cumulative function to a time series in which total and scaling operations are simultaneously performed point by point. Then, the cumulative features are used to characterize the change trend, which can be expressed as follows:

$$b_i^n = \frac{\sum_{j=1}^n f_i(j)}{\sqrt{\left|\sum_{j=1}^n f_i(j)\right|}} \quad i = 1, 2, \ldots I; \; n = 1, 2, \ldots N \tag{15}$$

where $b_i^n$ is the result of the cumulative change of $f_i(j)$ to the nth feature values, i is the number of feature kinds, n is the number of feature values, and $f_i(j)$ is the jth feature values of the ith feature kinds. As can be seen from Equation (15), if data preprocessing is not used, then the collected noise data will be gradually amplified with this cumulative operation, so it is necessary to reduce the noise effect by data preprocessing, as described above, before using the incremental transform algorithm.

### 4.4. Machining Trajectory Tracking Error Prediction Based on Transformer Modeling

The response of the machining trajectory at the current moment is closely related to the input data of several previous moments, which have a strong temporal correlation. When dealing with temporal correlation data, traditional deep learning algorithms, represented by recurrent neural networks (RNNs), can build predictive models by combining historical data [36]. Although RNNs have achieved good results in dealing with simple time series, they still suffer from issues with respect to the large consumption of computational resources and a severe loss of data features when dealing with nonlinear data such as machining trajectories, which involve a large amount of data, a comprehensive period, and complex features.

We adopt a transformer-based trajectory-tracking error prediction (TTTEP) model, which is a neural network based on an attention mechanism that can learn the input–output relationship of time series data and has achieved excellent results in many fields, such as natural language processing and computer vision [37]. The transformer model is distinct from other neural networks in that it not only considers the input of the current moment but also factors in the impact of past information on the current output. It can calculate the error value for a large amount of CNC machining trajectory data and is responsible for improving prediction accuracy and efficiency by accounting for the characteristics involved [38].

#### 4.4.1. Main Network Structure of TTTEP Model

The main network structure of the TTTEP model consists of an encoder and a decoder, where each layer of the encoder contains two sub-layers of the multi-head attention mechanism and a feed-forward neural network. Each decoder layer includes three sub-layers of the masked multi-head attention mechanism, the multi-head attention mechanism, and the feed-forward network. The main structure of the model is shown in Figure 10. When processing trajectory data containing multidimensional features are embedded into this model, the model can track the features in the learning data through temporal information due to the multi-head attention mechanism and stacked self-attention layer.

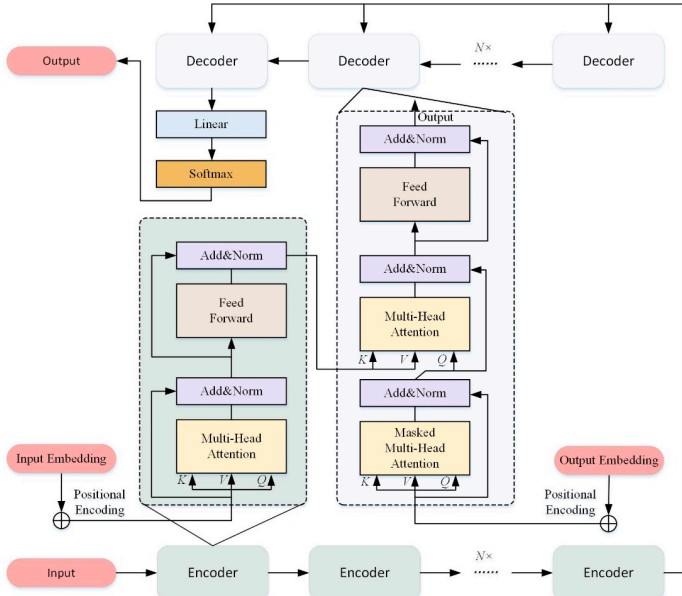

**Figure 10.** Main network structure of the TTTEP model.

Compared to traditional recurrent neural networks, the TTTEP model can better handle long-distance dependencies and avoid gradient vanishing and exploding problems in recursive structures. As a result, it performs better in data prediction tasks [37]. The multi-head attention mechanism and stacked self-attention layers in the encoder and decoder

blocks can effectively capture the long-distance dependencies and global information in the processed trajectory data. In addition to this, the model includes critical components such as residual connection and layer normalization. Residual connection, on the one hand, helps to alleviate the gradient vanishing problem in deep network training while facilitating information flow, which improves the model's training efficiency and generalization ability. In addition, layer normalization helps stabilize the training process, making the model more robust and reliable.

Overall, the TTTEP model has demonstrated excellent capabilities in handling machining trajectory data containing multidimensional features through its unique structure and mechanism. The model's main structures are described below.

### 4.4.2. Positional Encoding

Positional encoding assigns positional information to elements in the model's input sequence so that the model can distinguish between elements at different positions and capture the sequential relationships in the sequence. The strength of the model lies in the unique construction of its attention mechanism, but this results in it losing the ability to learn sequence position information. Therefore, in this paper, we encode the temporal and spatial properties of the processed trajectory sequences. The encoded objects embedded into the input sequences contain the embedded temporal position encoding and spatial position encoding vectors of the trajectory sequences and are obtained by

$$\zeta_{(i,n)} = t_{(i,n)} + S_n \tag{16}$$

$$S_n = \{P(pos, d)\}_{d=1}^{D} \tag{17}$$

The position information is encoded using the sin function and cos functions, and it is obtained by

$$P(pos, 2d) = \sin\left(\frac{pos}{10,000^{\frac{2d}{D}}}\right) \tag{18}$$

$$P(pos, 2d + 1) = \cos\left(\frac{pos}{10,000^{\frac{2d+1}{D}}}\right) \tag{19}$$

where pos is the position of the feature vector in the sequence and d denotes the feature vector's dimension.

Position encoding is essential for subsequent training because each dimension of the position encoding varies over time according to sinusoids of different frequencies, and the position encoding of each feature vector consists of sin and cos functions of different frequencies.

The transformer model uses dimensionally consistent word vectors as input features and label data, and this same dimensionality helps us to compute the encoding–decoding multi-head attention mechanism. However, the input features are multidimensional data in the processing trajectory error prediction scenario. In addition, the labels are one-dimensional uniaxial error values, resulting in a dimensional mismatch between the input features and labeled data, rendering the encoding–decoding multi-head attention mechanism unable to operate.

In this paper, we adopt the method introduced in Section 4.3, which directly uses the cumulative feature sequences as input at the encoder input. Meanwhile, a fully connected network layer is added at the decoder input and output for upscaling and downscaling operations at the input and output. With such an adjustment, the conflict or inconsistency between the dimensionality of the input features and the labeled data in the transformer-based processing trajectory following the error prediction model is resolved.

### 4.4.3. Encoder and Decoder

The encoder and decoder are the two core modules of the model. The encoder is responsible for encoding the input processing trajectory feature sequences and mapping

them onto intermediate vectors containing the processing trajectory features' information. The core principle of the encoder is its self-attention mechanism, which is mainly used to characterize correlations by calculating the similarity between the feature vectors to solve the long-range dependency capture problem. The purpose of the self-attention mechanism is to filter out a small amount of important information from the input sequence of the processing trajectory features and use weights to represent the importance of the information, allowing the model to focus on the more important information.

The essence of the self-attention mechanism is an addressing process, and scaled dot product attention is used to calculate the attention value of the feature matrix. Firstly, the correlation is calculated via the query matrix and key matrix's dot product. The weight coefficients are calculated via softmax normalization. The softmax function is a commonly used activation function, typically employed in multi-class classification problems to compute the probability distribution of each class. It is defined by the following formula:

$$\text{softmax}(x_i) = \frac{e^{x_i}}{\sum_j e^{x_j}} \tag{20}$$

where $x_i$ is an element in the vector, and $\sum_j e^{x_j}$ is the sum of all elements in the vector after applying the exponential function.

Finally, the value matrix is weighted and summed according to the weight coefficients, which can be obtained as

$$\begin{cases} Q = W^q X_n \\ K = W^k X_n \\ V = W^v X_n \end{cases} \tag{21}$$

$$\text{attention}(Q, K, V) = \text{softmax}\left(\frac{QK^T}{\sqrt{d}}\right) \cdot V \tag{22}$$

where $Q, K, V$ are the query matrix, key matrix, and value matrix, respectively, obtained by multiplying the input feature matrix $X_n$ by the corresponding weight matrix. $d$ is the dimension of the matrix $Q, K, V$.

The self-attention mechanism can help the model focus on crucial information in the input sequence. However, it can only learn relevant details in one representation space, so a single attention mechanism has limitations. In order to more fully synthesize the information in a processing trajectory, we use a multi-head self-attention mechanism that allows the model to simultaneously focus on information from different representational subspaces at various locations.

The multi-head self-attention mechanism is essentially a parallel application of multiple self-attention mechanisms, where each self-attention head focuses on learning information from different representational subspaces and finally splices and linearly transforms multiple attention values to obtain its final attention value, which helps to better generate the potential features present in the feature-complex processed trajectory data, which can be expressed as follows:

$$\begin{cases} \text{MultiHead}(Q, K, V) = \text{concat}(\text{head}_1, \text{head}_2, \dots, \text{head}_m) \cdot W \\ \text{head}_i = \text{attention}(QW_i^Q, KW_i^K, VW_i^V) \end{cases} \tag{23}$$

where $W$ is the multi-head attention weight matrix, $m$ is the number of attention heads, $W_i^Q, W_i^K, W_i^V$ are the weight matrices of the ith attention head $Q, K, V$, and $\text{head}_i$ is the computation result of the ith attention head. The concatenation result of the concat function is the merging of the calculations from each attention head. The definition of the attention function is given by Equation (22).

The decoder is responsible for decoding the intermediate vectors output from the encoder into an output sequence, and its core principles are a masking multi-head self-attention mechanism and a decoder–encoder multi-head attention mechanism. During

model training, decoding each time step of the sequence will be performed simultaneously due to the parallel computing nature of the decoder. This will lead to the computation of each time step learning the future labeled data information, which is not consistent with the real world.

Therefore, we add a masking operation to the decoder's multi-head self-attention mechanism to shield future labeled data information. Specifically, we introduce a matrix with a lower triangle and diagonal of 1 and an upper triangle of 0, which is multiplied by $QK^T$ during the computation of scaled dot product attention, causing the future sequence information to be set to 0. This ensures that the model learns information from previous and current moments.

In order to improve the accuracy of the transformer-model-based processing trajectory following the error prediction model, we introduce an encoding–decoding multi-head attention mechanism. The mechanism uses a query matrix Q from the output of the masking multi-head self-attention mechanism module containing information about the labeled data and a key matrix K with a value matrix V from the encoder output containing information about the input sequence. This allows for integrated learning of the dependencies between input feature vectors, the dependencies between labeled data, and the dependencies between the two.

After the self-attention mechanism, spatial changes must be performed using a feed-forward network (FFN). The FFN contains two linear transformation layers and the ReLu activation function [39], which transforms the spatial dimensions of the attentional outputs through an activation function, thus increasing the expressive power of the model. The computational formula is as follows:

$$FFN(x) = ReLU(xW_1 + b_1)W_2 + b_2 \tag{24}$$

where $W_1$, $W_2$ is the weight matrix of FFN.

In addition to the above, the increase in network depth will impact the accuracy of the processing trajectory tracking error prediction. We solve the problem of network degradation to a large extent by adding a residual connection operation between each sub-layer of the encoder and the decoder, which deals with the data at the connection through normalization. In addition, in order to increase the training speed and improve the stability of the training, each sublayer uses a layer normalization operation, which can be obtained as follows:

$$b = LayerNorm\,(a + Sublayer(a)) \tag{25}$$

where a and b are the network's input and output parameters, Sublayer(a) is the processing function for the attention mechanism layer and FFN, and LayerNorm is the function that normalizes all the hidden units of each sample to prevent overfitting and improve the robustness of the model.

### 4.5. Adaptive Error Compensation Methods

In the CNC machining process, the interpolator in the CNC system will input the machining trajectory information, which will be transformed into a series of data points with specific intervals. These data points will be discretized according to their interpolation period. Since the interpolation period is usually very short, in most cases within 2 ms, the interval between the discrete interpolation points is also tiny, and the bow height error is almost negligible. After interpolation, the output data point sequence is regarded as the theoretical trajectory. However, the actual situation is often complex, and processing errors are inevitable, requiring compensation for the actual trajectory.

Based on Equation (11) in Section 4.1 and the tracking error of the model output in Section 4.3, the actual position point location information can be calculated. We propose an adaptive error compensation method to find the machining trajectory error, calculate the compensation length, and compensate the position using an adaptive method. This

method is mainly divided into two steps; the first step involves confirming the closest distance of the actual position point to the ideal machining trajectory of the pendant point Po, and the second step relates to the pendant point's distance along the average direction of the inverse calculation of the interpolation point Pc. Finally, according to the position information of the interpolated point, after the compensation of the G-code is changed, the curve information of the actual machining trajectory is infinitely close to that of the commanded trajectory. The proximity point of the actual position to the command position is determined by defining a sliding-window-based proximity search, as shown in Figure 11. Compensate the interpolation points of the machining trajectories arranged in chronological order according to the direction pointed by the arrow.

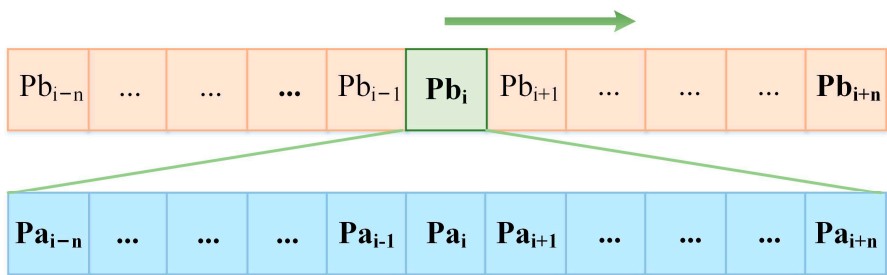

**Figure 11.** Proximity point search based on a sliding window approach.

The user defines the length m of the sliding window. The distance from the theoretical command position point $Pa_{i-n}$ to the nearest point in $Pa_{i+n}$ and to the actual position point $Pb_i$ can be obtained as follows:

$$l = \min || \overrightarrow{Pb_i Pa_m} ||, \ m \in (i-n, \ i+n) \tag{26}$$

After finding the nearest point $Pa_m$ according to Equation (26), it can be determined that the vertical foot point Po is on the line segment $\overrightarrow{Pa_{m-1}Pa_m}$ or $\overrightarrow{Pa_m Pa_{m+1}}$. The distance of the actual position point $Pb_i$ to the line segment $\overrightarrow{Pa_{k-1}Pa_k}$ or $\overrightarrow{Pa_{k-1}Pa_k}$ is calculated by Equation (10), taking the minimum value of the machining trajectory error $\varepsilon_\kappa$ and determining the vertical foot point Po. Then, the actual position point is placed along the vertical foot point Po and the normal line of the trajectory is moved twice the processing trajectory error distance $\varepsilon_\kappa$ to obtain the compensation point $Pc_k$, which can be expressed as follows:

$$Pc_i^\kappa = Pb_i^\kappa - 2\varepsilon_i^\kappa \cdot n_i^\kappa, \kappa = x, \ y, \ z \tag{27}$$

where $n_i$ is the normal vector of $Po_k$ on the trajectory. The adaptive error method is shown in Figure 12.

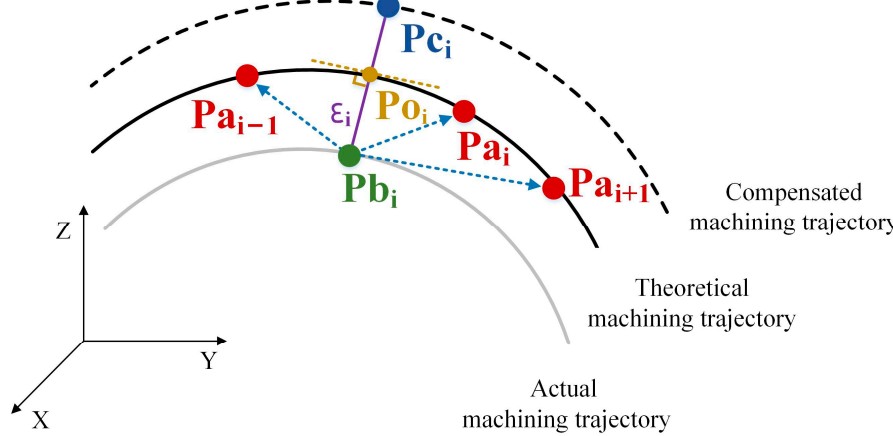

**Figure 12.** Adaptive error compensation for the machining trajectory.

## 5. Experiments and Results

### 5.1. Overall Strategy

The overall strategy of this machining trajectory prediction and compensation method is shown in Figure 13. By analyzing the trajectory data acquisition during actual machining, the error prediction and compensation method of the machining trajectory is realized under the digital twin of the CNC system based on a hybrid model. The specific steps are as follows:

1.　In the process of collecting theoretical command position and actual position data for the machining trajectories of the CNC system, it is essential to ensure the accuracy and stability of the data collection equipment. Real-time recording and collection of theoretical command position and actual position data generated by the CNC system during the machining process are carried out through precise sensors and measuring tools. These data may include the position coordinates of each axis, speed information, etc. Prior to processing the data, calibration and filtering are conducted to ensure the reliability and accuracy of the data.

2.　When training the AI algorithm model described in Section 4.4 on the machining trajectory data of the CNC system axes, preprocessing of the data is necessary. This includes data cleaning, feature extraction, and other steps to enhance the accuracy and generalization ability of the training model. By inputting any given reference position point, the AI algorithm model can learn and establish error models for each axis of the machining trajectories, enabling the prediction and calculation of errors. This establishes a crucial foundation and support for subsequent error compensation.

3.　When applying the adaptive error compensation method described in Section 4.5 to compensate for the original reference positions, adjustments are made based on the actual machining conditions and error prediction results to obtain new compensated reference positions. This process involves techniques such as parameter optimization and feedback control to ensure that the post-compensation reference positions effectively guide the motion trajectories of the CNC system, facilitating the accurate prediction and compensation of machining trajectory errors. The new reference positions are stored in the internal data buffer of the CNC system to provide a basis for real-time error compensation, thereby enhancing machining accuracy and efficiency.

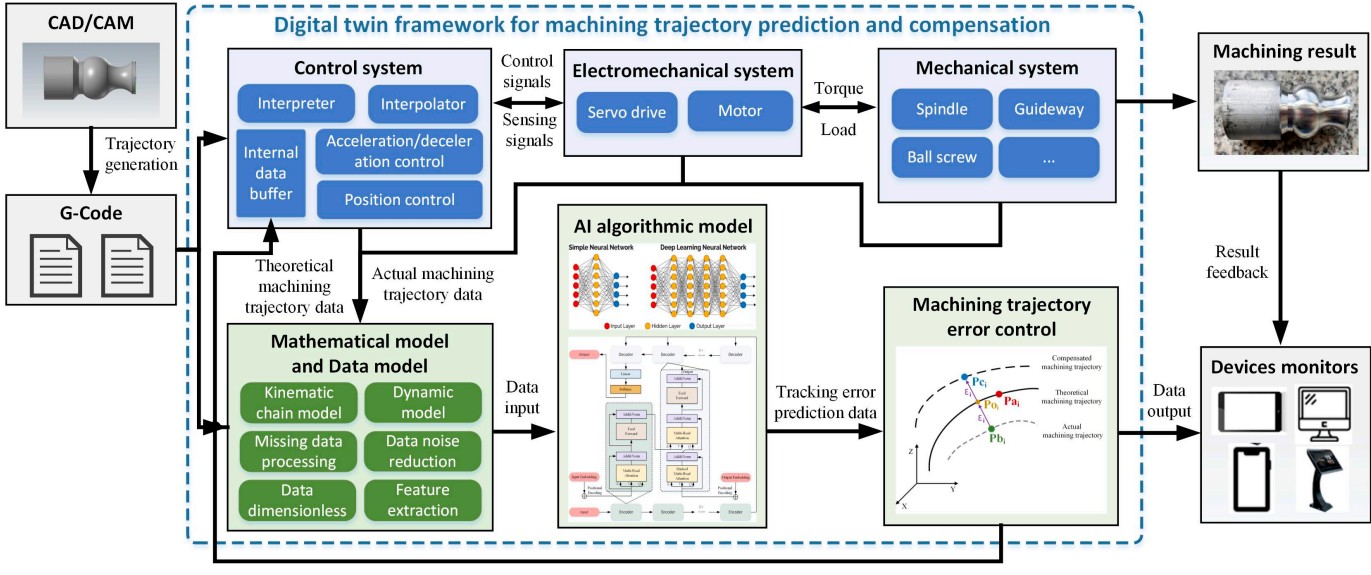

**Figure 13.** Overall strategy for machining trajectory error prediction and compensation.

By implementing the detailed steps and technical methods mentioned above, comprehensive management and control of the machining trajectory errors in the CNC system can be achieved, providing robust support and assurance for optimizing and improving the CNC machining process.

### 5.2. Digital Twin Implementation

Based on the above, we investigated a digital twin modeling implementation method for machining trajectory error prediction and compensation. The physical entity was a RuiFeng turn-milling compound machine tool, model 6152WY (Ruifeng Hardware Machinery Co., Ltd., Foshan, China), equipped with the Lantian CNC system GJ430 (Shenyang CASNC Technology Co., Ltd., Shenyang, China). In the virtual space, we used the CNC digital twin modeling method mentioned previously to model, locate, and render the CNC machine tool in three dimensions through 3DMAX (version number: 3ds Max 2022). We then developed a simultaneous monitoring system of the machining process using Unity3d (version number: 2021.2.10f1c1 Personal) and C# to establish the CNC's digital twin model, as shown in Figure 14. Python language was used to realize the machining trajectory error prediction and compensation algorithm model. The Oracle database was used to store and manage the machining trajectory data to complete the online optimization of process parameters during the machining process.

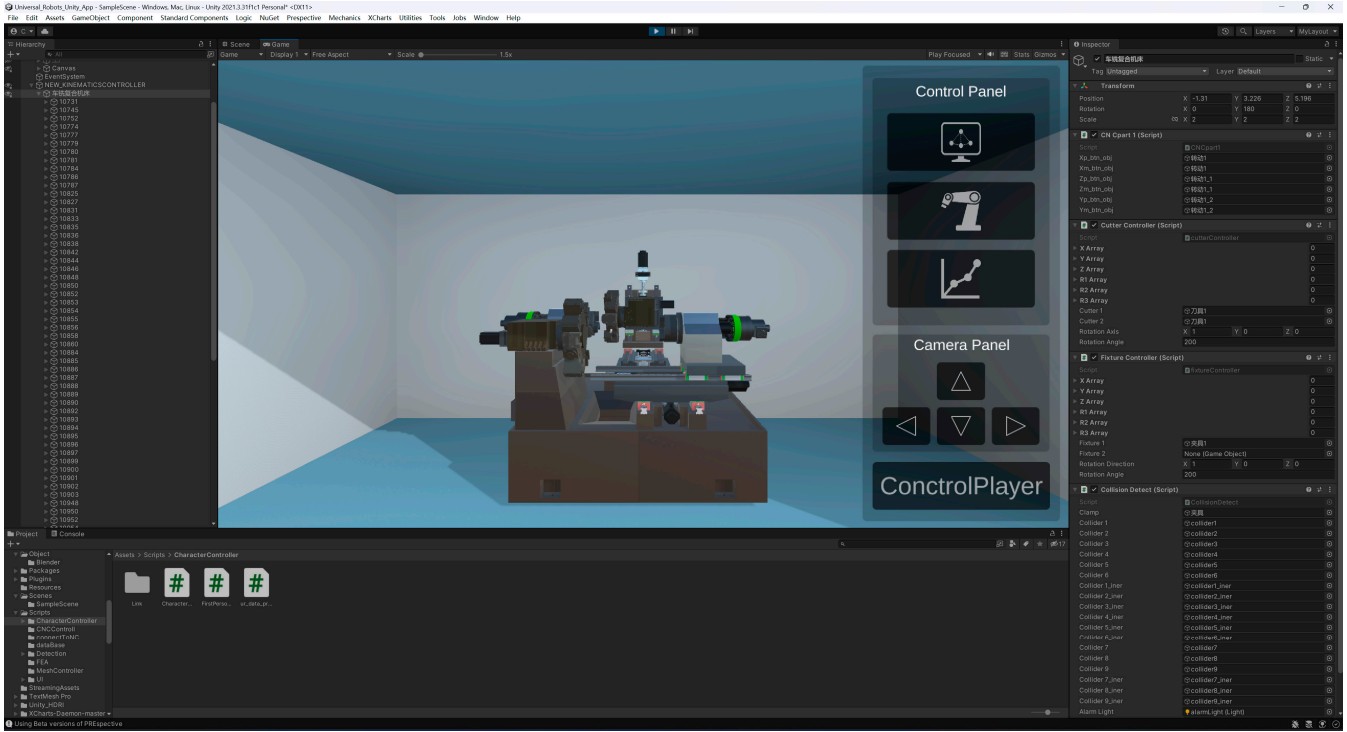

**Figure 14.** Visualization of the digital twin model of the CNC system.

In this study, we use the service-oriented OPC UA architecture to establish the digital twin virtual–reality synchronization interface, which realizes the data-driven synchronization of physical entities and virtual space, as shown in Figure 15. The virtual digitization of the CNC machining process was achieved through the simulation of the entire machining process from trajectory interpolation to cutting machine modeling.

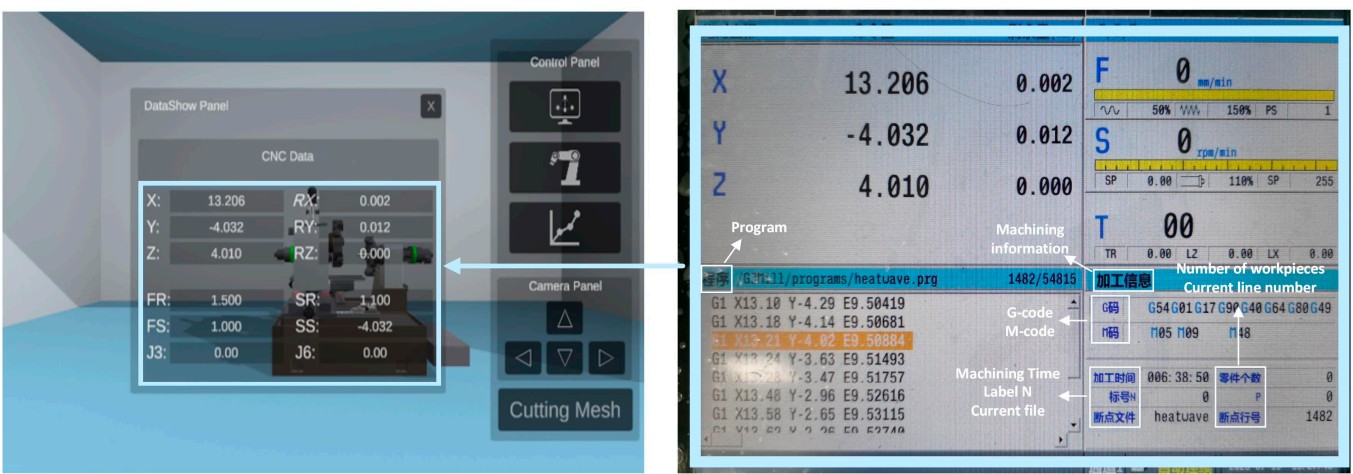

**Figure 15.** Data synchronization of physical entities and virtual spaces of the CNC system.

*5.3. Experimental Results of Machining Trajectory Error Prediction and Compensation*

In order to verify the effectiveness of the method proposed in this paper, relevant verification experiments were carried out on the machining tool (6152WY). First of all, it was necessary to collect machining trajectory data during the machining of a prototype on the machining tool, and the specific data collection method is shown in Figure 16.

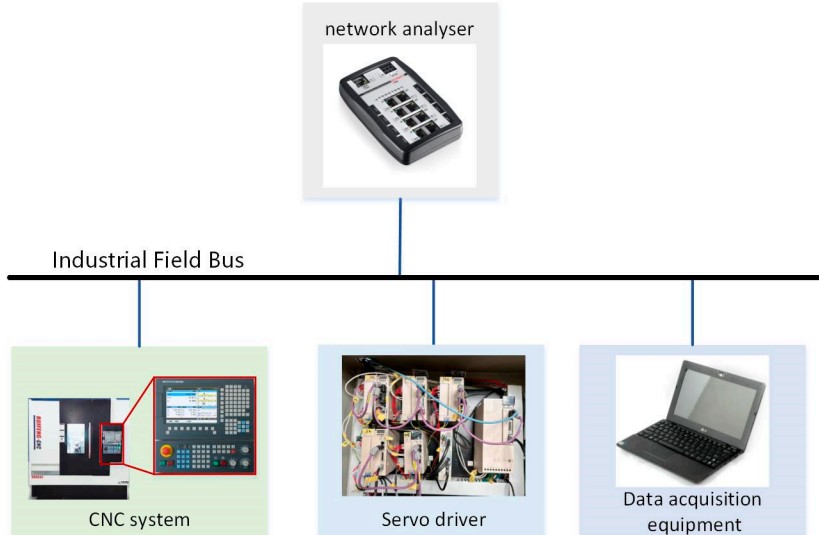

**Figure 16.** Machining trajectory data acquisition.

In our experimental environment, we used the Beckhoff ET2000 network analyzer for data acquisition based on the MECHATROLINK-III bus protocol communication. This system is capable of recording up to four independent channels synchronously at 100 Mbit/s without any limitation, and all frames in transmission (bi-directional) are timestamped with high accuracy (up to 1 ns) in the probe hardware, allowing for a precise time analysis of the connected network segments. Therefore, connecting it in parallel with the data acquisition device, the CNC and the SGDV servo drive allowed for a more accurate acquisition of the command position information from the CNC and the feedback position information from the servo drive. Finally, the WireShark software (version number: 4.1.0) packet grabber tool was used to obtain the machining trajectory data collected using the network analyzer, including command and feedback data information, and the sampling frequency was set to 2 ms. The experimental environment is shown in Figure 17.

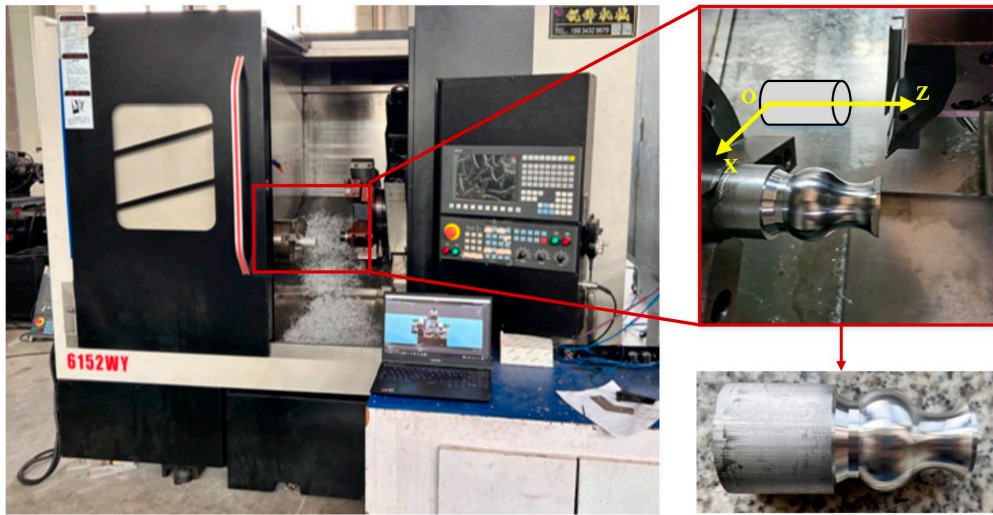

**Figure 17.** Experimental environment for predicting and compensating machining trajectory errors in CNC systems.

Validation tests were carried out using machined workpieces with a combination of straight lines and circular arcs, with a maximum feed rate of 1500 mm/min. The experimental results of the machining trajectory path, including the command position of the CNC system and the feedback position of the servo drive, are shown in Figure 18. At the same time, in order to ensure the robustness of the introduced method, this paper divides the collected machining trajectory into two parts for verification. Trajectory 1 mainly contains the tool path of the arc, and trajectory 2 is the tool path of the combination of the line and arc with a corner.

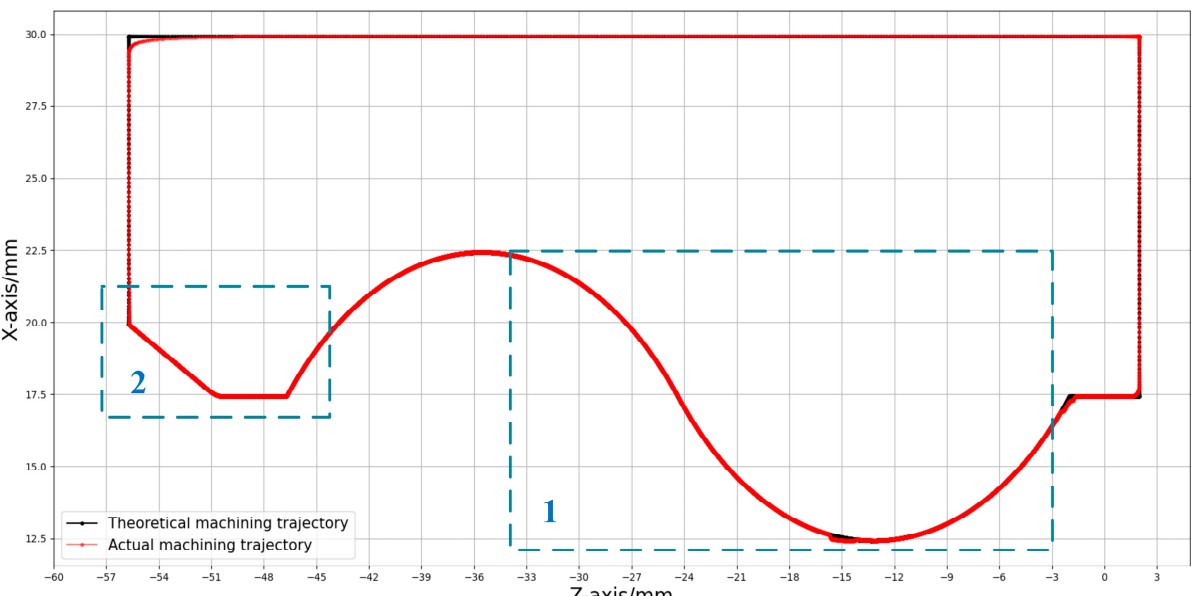

**Figure 18.** Theoretical machining trajectory and actual machining trajectory used in the experiments.

Before predicting and compensating for the machining trajectory errors, the acquired trajectory data were processed as in Section 4.2 and then divided into a training set and a test set. The features of the training set were processed using the method in Section 4.3 to ensure that the data were in the same scale range and to eliminate the differences in magnitude between the different features, which can help improve the robustness and accuracy of the model. The final processing results were used as inputs to the transformer model, and the

tracking errors were used as training labels for the model. The trained model's mean square error (MSE) was calculated as its loss function, and the Adam optimization algorithm was used for model training and optimization.

We input the data from the test set into the trained transformer model, predicted the trajectory tracking error data, and compared these with the actual tracking errors to verify the model's effectiveness. The tracking error was also used as one of the data sources for the follow-up error, which provided data support for the machining performance improvement of the CNC system.

The algorithmic network was trained on a computer configured with an NVIDIA GeForce RTX 2060 GPU(NVIDIA Corporation, Santa Clara, CA, USA), where the deep learning framework was PyTorch version 1.8 and CUDA version 11.1. In the actual processing trajectory data captured, the training data accounted for 67% of the data, and the test data accounted for 33%. The sequence length was set at T = 30, the frequency threshold of the error compensation coefficient was 1 HZ, and the deflation coefficient was 0.01. The Adam optimizer was used in the model training process, and the learning rate was 0.005. After 100 rounds of training, the loss function for predicting machining trajectory errors in the *X*-axis and *Z*-axis of the motion axes was determined, and the results are shown in Figure 19.

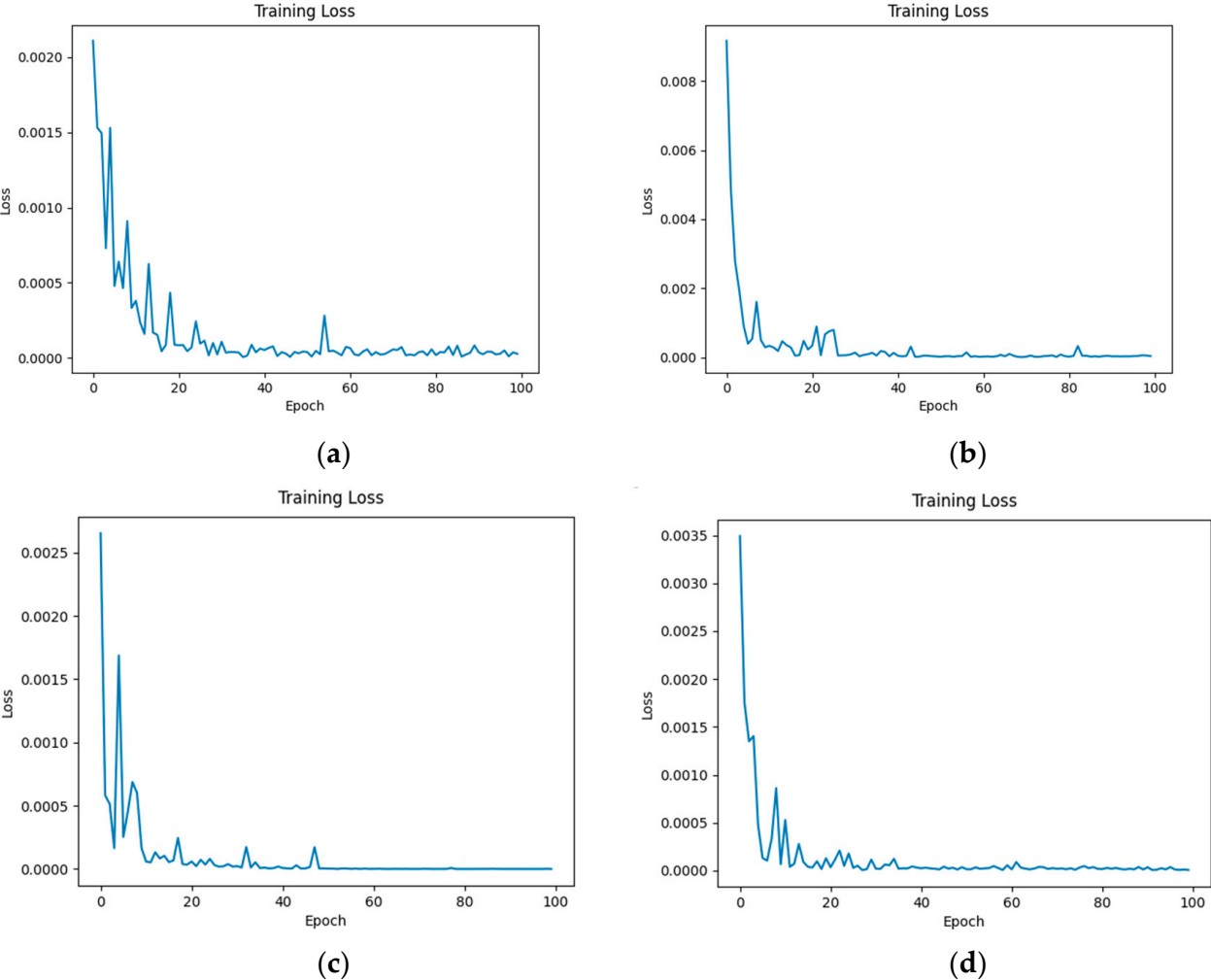

**Figure 19.** (**a**) The loss function of the *X*-axis tracking error prediction results for trajectory 1; (**b**) the loss function of the *Z*-axis tracking error prediction results for trajectory 1; (**c**) the loss function of the *X*-axis tracking error prediction results for trajectory 2; and (**d**) the loss function of the *X*-axis tracking error prediction results for trajectory 2.

The experimental results of the *X*-axis and *Z*-axis machining trajectory tracking error predictions for the machine are shown in Figure 20. The results show that the predicted results are basically in agreement with the actual collected tracking error.

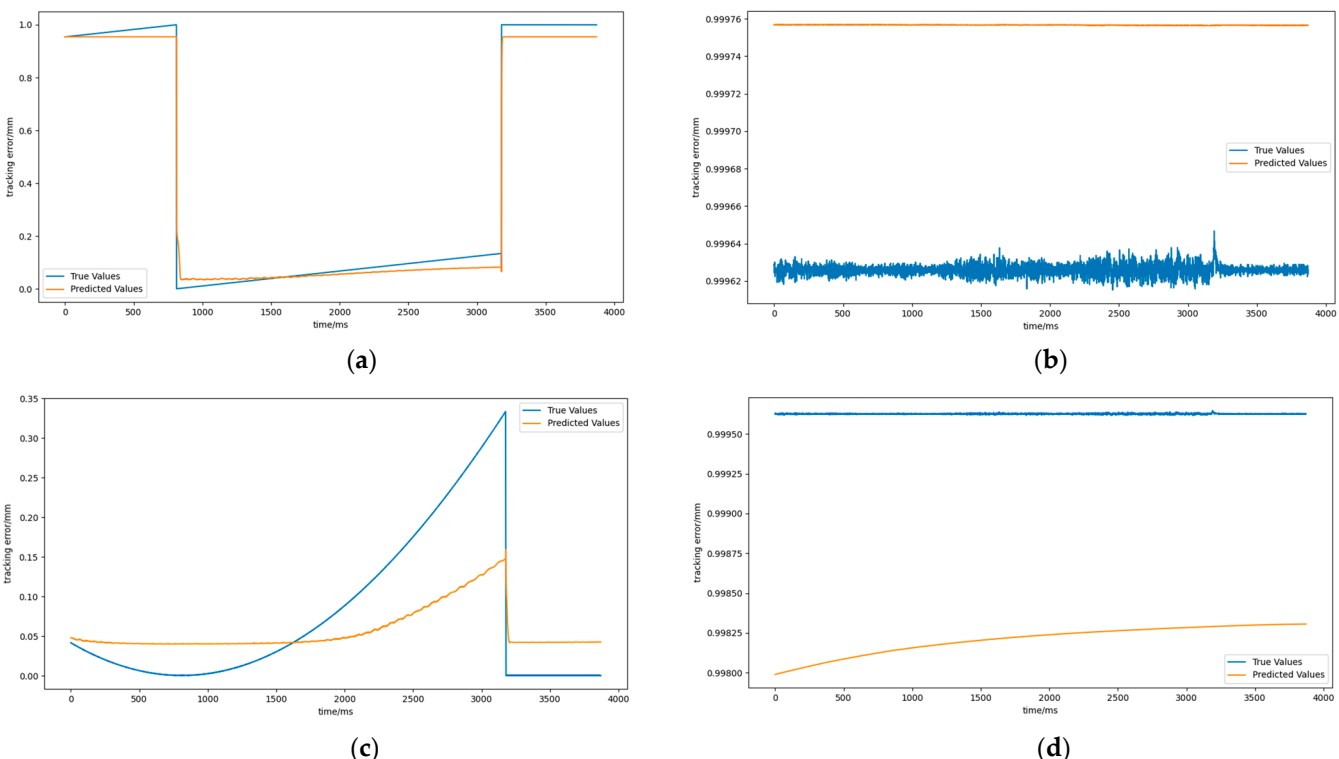

**Figure 20.** (**a**) Predicted and true values of *X*-axis tracking error for trajectory 1; (**b**) predicted and true values of *Z*-axis tracking error for trajectory 1; (**c**) predicted and true values of *X*-axis tracking error for trajectory 2; and (**d**) predicted and true values of *Z*-axis tracking error for trajectory 2.

In order to compare the relationship between the predicted and actual results in more detail, the difference between the true values and the predicted values was calculated. The results are shown in Figure 21. The maximum value of the *X*-axis tracking error prediction is 77.94 μm for trajectory 1. The maximum value of the *Z*-axis tracking error prediction is 0.34 μm for trajectory 1. The maximum value of the *X*-axis tracking error prediction is 0.96 μm for trajectory 2. The maximum value of the *Z*-axis tracking error prediction is 0.0076 μm for trajectory 2. By analyzing the machining process and technology of the parts, it can be seen that the reason for this situation is that the *Z*-axis machining trajectory is mostly a straight line. In contrast, the *X*-axis movement contains a large number of corners and arcs, so there is a larger error prediction error at the point of inflection. Still, this does not have much of an effect on the overall prediction results.

Currently, in the time series data analysis field, the mainstream neural networks used include SVM and LSTM. Table 1 presents the trajectory data for the collected *X*-axis and *Z*-axis, along with the prediction accuracy without using the data processing method described in Section 4.2. As shown in Table 1, the model's prediction accuracy is higher when the data processing method is employed.

A comparison of the prediction method proposed in this paper with other time-series-prediction-based models is shown in Table 2. The results show that our method based on the transformer model is a little slower than the traditional algorithms to produce results, but its results are the most accurate.

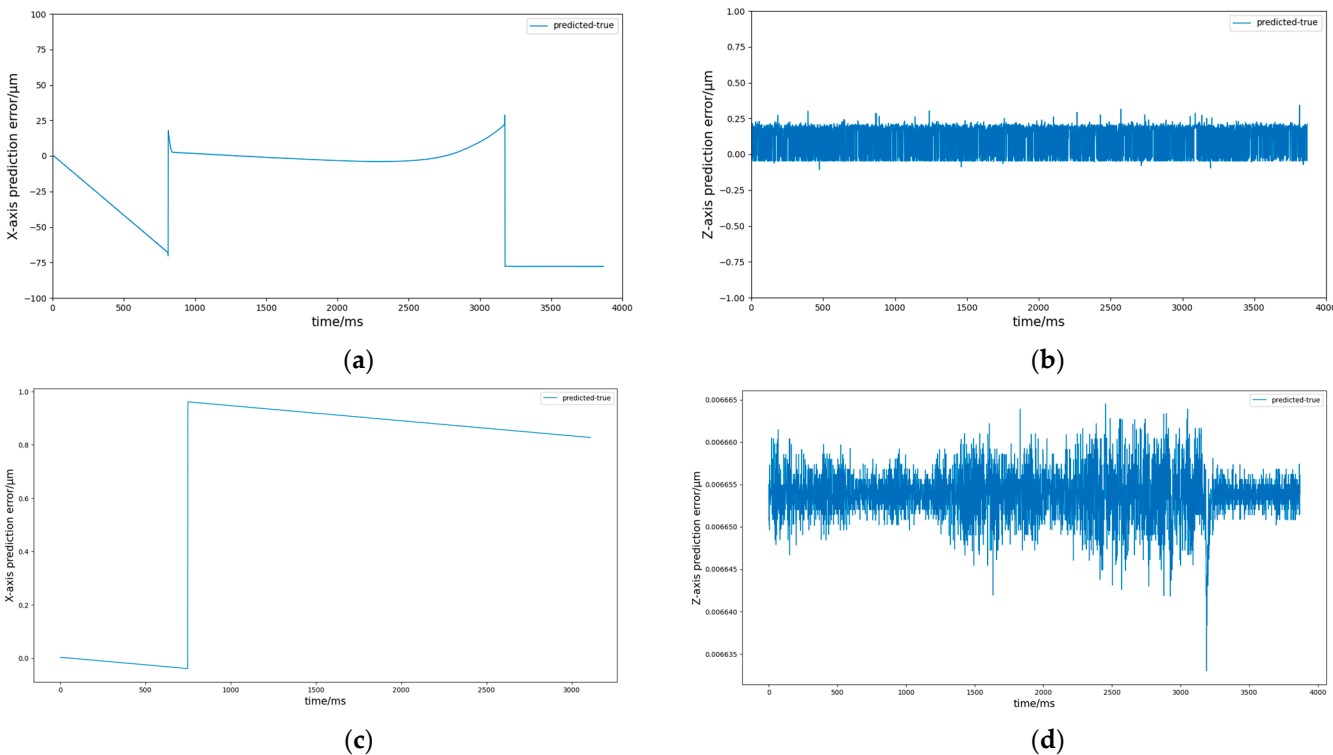

**Figure 21.** (**a**) Prediction error of *X*-axis tracking error for trajectory 1; (**b**) prediction error of *Z*-axis tracking error for trajectory 1; (**c**) prediction error of *X*-axis tracking error for trajectory 2; and (**d**) prediction error of *Z*-axis tracking error for trajectory 2.

**Table 1.** Prediction accuracy using data processing methods and using no data processing methods.

| Neural Network | Prediction Accuracy (%) | | | |
| --- | --- | --- | --- | --- |
| | Using No Data Processing | | Using Data Processing | |
| | *X*-Axis | *Z*-Axis | *X*-Axis | *Z*-Axis |
| SVM | 69.67 | 77.19 | 95.12 | 97.19 |
| LSTM | 77.19 | 78.63 | 97.68 | 98.14 |
| Transformer | 73.14 | 79.34 | 98.75 | 98.81 |

**Table 2.** Comparison of experimental results of different machining trajectory error prediction models.

| Neural Network | Maximum Prediction Error (μm) | | Time (s) | |
| --- | --- | --- | --- | --- |
| | *X*-Axis | *Z*-Axis | *X*-Axis | *Z*-Axis |
| SVM | 123.45 | 0.98 | 16.54 | 17.14 |
| LSTM | 96.51 | 0.63 | 26.45 | 28.44 |
| Transformer | 77.94 | 0.34 | 25.14 | 26.12 |

After obtaining these prediction results, the error was compensated for, using the adaptive error compensation method in Section 4.5 for trajectory 1 and trajectory 2. The results are shown in Figure 22.

In trajectory 1, the maximum value is 0.169 mm and the mean value is 0.096 mm before compensation using our method. The maximum value of the error is reduced to 0.082 mm and the mean value is reduced to 0.041 mm after the compensation. The maximum value of the error of the machining trajectory is reduced by 51.47%, and the mean value is reduced by 57.29%. In trajectory 2, the maximum value is 0.173 mm and the mean value is 0.081 mm before compensation using our method. The maximum value of the error is reduced

to 0.083 mm and the mean value is reduced to 0.033 mm after the compensation. The maximum value of the error of the machining trajectory is reduced by 52.02%, and the mean value is reduced by 59.26%. The above results show the feasibility and effectiveness of the method presented in this paper.

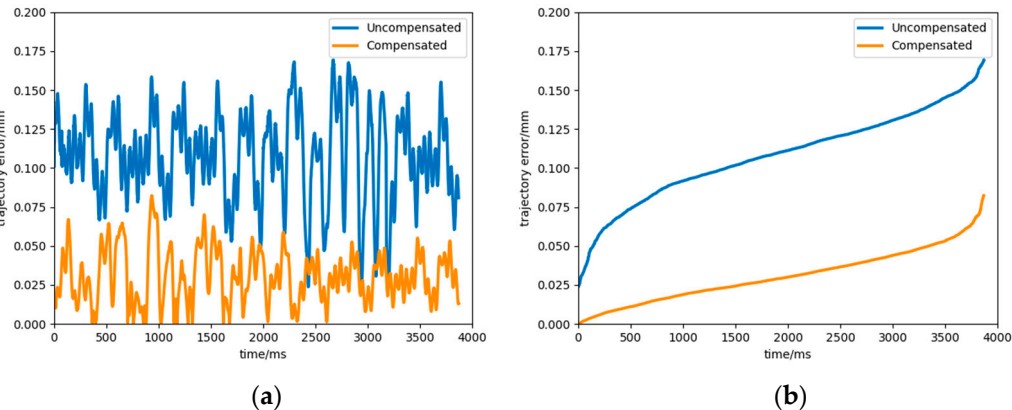

**Figure 22.** (**a**) Comparison of uncompensated and compensated for errors of machining trajectory 1; (**b**) Comparison of uncompensated and compensated for errors of machining trajectory 2.

## 6. Conclusions

The advancement of intelligent manufacturing has created new requirements for CNC systems. A digital twin combines data sensing, big data processing, and AI algorithms to model and analyze the machining process. This approach provides a new direction for the intelligent and efficient machining of CNC systems, which can effectively fulfill these new demands.

In this study, we analyzed the structure of a CNC system and constructed a digital twin framework of the CNC system based on a hybrid model. To improve the accuracy and efficiency of the CNC system, we analyzed its machining trajectory data using multi-dimensional data combined with the mechanism of machining trajectory error prediction and compensation. By fully mining and utilizing the relevant features of the machining trajectory sequence data, the machining trajectory following an error prediction algorithm based on the transformer model that was used to predict the machining trajectory error. Based on the prediction results, the adaptive error compensation method compensates for the error, ensuring the subsequent workpiece machining's accuracy and quality and resulting in subsequent high-speed, high-precision machining.

Finally, the case of digital-twin-driven machining trajectory error prediction and compensation was presented to verify the feasibility and effectiveness of this method through specific experiments. This study provides a new reference for improving the machining performance of CNC systems and solving the problem of machining trajectory error prediction and compensation when developing CNC systems towards intelligence.

## 7. Discussion

### 7.1. Limitations

Based on digital twin theory, this study focused on digital-twin-driven CNC machining trajectory error prediction and compensation methods by integrating digital twin modeling technology and AI algorithms. However, the CNC system involves other vital technologies (such as multi-axis linkage control, vibration suppression surface machining optimization, etc.) in addition to the above motion control technologies, and these still need to be studied. In the production process, how to carry out the profound combination of CNC systems and digital twin technology still requires further consideration.

*7.2. Future Research*

Digital twin technology provides a new approach to realizing the intelligence of CNC systems. With the increase in external devices such as sensors, the amount and dimensions of data generated by CNC systems during the machining process are increasing and have typical big data characteristics. Future research directions will involve exploring big data technologies such as data sensing, data mining, distributed computing, and big data storage for control strategies such as quality enhancement, process optimization, fault diagnosis, and production management, organically combining monitoring, prediction, and optimization functions based on digital twin technology to achieve the data-driven online dynamic optimization of predicted process parameters. These studies will further enhance the intelligence of the CNC system and provide more comprehensive digital support for its design, production, products, and services.

**Author Contributions:** Conceptualization, W.H., L.Z. and D.Y.; methodology, W.H. and L.Z.; software W.H.; validation, W.H., Y.H. and Z.Z.; formal analysis, W.H.; investigation, Z.Z. and Y.Q.; resources, W.H. and L.Z.; data curation, W.H. and Z.Z.; writing—original draft preparation, W.H. and Z.Z.; writing—review and editing, W.H.; visualization, W.H. and Y.Q.; supervision, Y.H. and D.Y.; project administration, D.Y.; funding acquisition, D.Y. All authors have read and agreed to the published version of the manuscript.

**Funding:** This research was funded by the Quality and Reliability Testing and Evaluation Service Platform for Industrial Machine Tool in China (Grant No. 2022-232-223-01).

**Data Availability Statement:** The datasets generated and analyzed during the current study are available from the corresponding authors upon reasonable request.

**Conflicts of Interest:** Author Yi Hu was employed by the company Shenyang CASNC Technology Co., Ltd. The remaining authors declare that the research was conducted in the absence of any commercial or financial relationships that could be construed as a potential conflict of interest.

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
