# Peer review of "A Hybrid-Model-Based CNC Machining Trajectory Error Prediction and Compensation Method"

_electronics, doi:10.3390/electronics13061143_

Round 1

Reviewer 1 Report

Comments and Suggestions for Authors

Review of electronics-2869544-peer-review-v1: “A hybrid model-based for CNC machining trajectory error prediction and compensation method”

The subject of the paper is relevant with the topics of the journal.

The references are well selected and used within the text. At the same time, they are mainly selected from researches performed the last five years.

The paper’s contribution is well stated and at the end supported by the results.

The paper gradually develops the authors’ contribution and is very well structured.

It proposes a hybrid model-based machining trajectory error prediction and compensation method to address these issues. Use of a digital twin framework of the CNC system based on a hybrid model is built and supported by detail explanation of the work completed.

Both limitations and future work make complete the research presented.

I have enjoyed reading it and look forward to reading more of the authors’ work in the future.

My proposal to the editors is to accept as is.

Author Response

Thank you very much for your review and positive feedback on the paper. We are glad to receive your recognition and support. We will continue to strive to make more contributions to the field of research.

We also appreciate your mention of the limitations and suggestions for future work in the paper. We will further explore and improve upon these issues in our future research. Once again, thank you for your valuable review and recommendations. We look forward to presenting even better research outcomes for you in the future.

Reviewer 2 Report

Comments and Suggestions for Authors

This article presents an interesting model proposal applied to the prediction and compensation of CNC machining trajectory errors. In general, the manuscript is well structured, the theoretical development of the algorithm is presented, as well as the experimentation in an appropriate manner to validate the proposal. However, I consider that there are some aspects to consider before a possible publication:

1. Improve image quality 5 and 8. Simplify the information with the aim of making the image more readable.

2. In section 4, several functions are mentioned such as "softmax", "concact" and "attention", among others. I recommend clarifying how these functions are defined.

3. Conventional G code is mainly based on the interpolation of trajectories with lines and arcs (G01 and G02). However, the implementation of modern interpolation codes based on spline curves (G05) improves machine performance and reduces tracking error. In his experimentation it is mentioned that free curves were used (line 751 of the document), are these free curves splines or which curves are you referring to?

4. In the sense of the previous comment, an important element of CNC machining performance is the design of movement profiles, that is, the control over speed, acceleration and jerk, since these parameters directly influence the dynamics of the machine and contribute to reduce tracking error. Does the CNC controller used have versatility over these parameters?

5. A Beckhoff network analyzer is used in the experimentation, it is mentioned that it has high processing and communication speed. However, the error prediction and compensation model is developed on the PC, were there no communication or latency problems between the CNC and the PC?

6. The noise reduction and missing data are important parts of the proposed methodology, but results of their performance in the experimentation are not shown. I recommend showing or briefly mentioning the results obtained at this stage.

7. Initially it is commented that the CNC system generates different machining information such as position, speed, acceleration. But the methodology does not describe how all this data was processed, it only emphasizes the position read from the encoder. How do the other variables influence?

Comments on the Quality of English Language

Overall, the document is well written. However, I recommend a complete review to avoid any writing errors.

Author Response

Response to Reviewer 2 Comments

We feel great thanks for your professional review work on our article. As you are concerned, there are several problems that need to be addressed. According to your nice suggestions, we have made extensive corrections to our previous manuscript. The detailed point-by-point responses are listed below.

Point 1: Improve image quality 5 and 8. Simplify the information with the aim of making the image more readable.

Response 1: Thanks for your kind reminder. We have improved the quality of  Figure 5 [Pg9] and Figure 8[Pg12], and simplified the information in the figures to enhance their readability.

Point 2: In section 4, several functions are mentioned such as "softmax", "concact" and "attention", among others. I recommend clarifying how these functions are defined.

Response 2: Thank you for the precious comment. According to your suggestion, we supplemented the definitions of these concepts. We made the following modifications and additions to the definition of  "softmax"[Pg18, Line623-629], "concat" [Pg18, Line650-651] and "attention" [Pg19, Line651-652] as follow:

"The softmax function is a commonly used activation function, typically employed in multi-class classification problems to compute the probability distribution for each class. It is defined by the following formula:

(20)

Where  is an element in the vector, and  is the sum of all elements in the vector after applying the exponential function. "

"The concatenation result of the concat function is the merging of the calculations from each attention head. "

"The definition of the attention function is given by equation 22. "

And the "concact" in the paper is a spelling error, and we have corrected it to "concat".

Point 3: Conventional G code is mainly based on the interpolation of trajectories with lines and arcs (G01 and G02). However, the implementation of modern interpolation codes based on spline curves (G05) improves machine performance and reduces tracking error. In his experimentation it is mentioned that free curves were used (line 751 of the document), are these free curves splines or which curves are you referring to?

Response 3: Thank you very much for the question. In this paper, the G code used in our experiments mainly consists of straight lines and arcs, without other curves. We apologize for any confusion caused by our previous imprecise description and have removed the reference to other curves in the text to make it more accurate. We revised the sentence as follows[Pg24, Line808-809]:

" Validation tests were carried out using machined workpieces with a combination of straight lines and circular arcs, with a maximum feed rate of 1500 mm/min. "

Point 4: In the sense of the previous comment, an important element of CNC machining performance is the design of movement profiles, that is, the control over speed, acceleration and jerk, since these parameters directly influence the dynamics of the machine and contribute to reduce tracking error. Does the CNC controller used have versatility over these parameters?

Response 4: Thank you very much for the question. The CNC system we used has the capability to control speed, acceleration, and jerk, with these functions processed in the feedforward control loop based on the G code file. Tracking error is a key factor affecting numerical control machining performance. Therefore, this paper focuses on studying position control within tracking errors and implementing compensation for machining trajectories. This paper treats speed, acceleration, and jerk as some characteristics of position compensation, and their control requires separate control algorithms for implementation.

Point 5: A Beckhoff network analyzer is used in the experimentation, it is mentioned that it has high processing and communication speed. However, the error prediction and compensation model is developed on the PC, were there no communication or latency problems between the CNC and the PC?

Response 5: Thank you very much for the question. As you mentioned, the communication delay between devices is crucial in the experiment. The Beckhoff network analyzer used in this study is the ET2000 model, which provides high-precision timestamping for data frames (bi-directional transmission) with a resolution of 1 ns, enabling extremely accurate time analysis of the connected network segments. According to the product specifications of the device (https://www.beckhoff.com.cn/zh-cn/products/i-o/ethercat-development-products/elxxxx-etxxxx-fbxxxx-hardware/et2000.html), its low cycle delay of 1 µs results in minimal impact on the system.

In the experimental process, we observed that the communication delay between the CNC and PC during data acquisition is very small (≤2ms), and therefore can be considered negligible. Furthermore, the data acquisition method employed in this study is based on the actual machining process of the machine tool. The error prediction and compensation model are trained offline, and the latency between the Beckhoff network analyzer and the PC is not a significant concern when using this offline model during the machining process.

On one hand, the latency from the digital twin device to the physical device is minimal, and on the other hand, because the trajectory error prediction and compensation method in this study is conducted offline, the communication or latency issues between the CNC and PC have minimal impact on the experimental results presented in this paper.

Point 6: The noise reduction and missing data are important parts of the proposed methodology, but results of their performance in the experimentation are not shown. I recommend showing or briefly mentioning the results obtained at this stage.

Response 6: Thank you for your suggestion. Based on your suggestion, we conducted experiments using both the raw data without noise reduction and missing data processing, and the processed data. We have added Table 1 as part of the experimental results to demonstrate the performance improvement achieved by applying the data processing method. Additionally, we provided textual explanations for the experimental results obtained from dataset. The modifications and additions are as follows [Pg27, Line870-875]:

" Table 1 presents the trajectory data for the collected X-axis and Z-axis, along with the prediction accuracy without using the data processing method described in Section 4.2. As shown in Table 1, the model's prediction accuracy is higher when the data processing method is employed.

Table 1. Prediction accuracy using data processing methods and unsuitable data processing methods. "

Point 7: Initially it is commented that the CNC system generates different machining information such as position, speed, acceleration. But the methodology does not describe how all this data was processed, it only emphasizes the position read from the encoder. How do the other variables influence?

Response 7: Thank you very much for the question. In our study, the CNC system generated various machining information variables such as position, speed, and acceleration, which play a role in estimating and compensating for trajectory errors. While speed and acceleration do affect machining performance, compensating for these variables is not the focus of our study. Therefore, in our current work, we use these variables as features for training, with a focus on the impact of position errors in the machining trajectory on performance.

Reviewer 3 Report

Comments and Suggestions for Authors

The manuscript presents a method for trajectory error prediction/compensation. the manuscript needs to be revised to a wide extent prior to acceptance. 

The manuscript introduces to a large extent the principles behind the framework that is presented but fails to discuss the specific details of the model and how it was developed, making the results not reproducible.

Details on the toolpaths tested, the result of the machining process and the measured deviations from the nominal profile programmed are entirely missing.

The findings of the model need to be verified with multiple toolpaths to ensure the robustness of the approach presented.

Comments on the Quality of English Language

The text contains a lot of syntactical errors and needs to be extensivelly revised and proofred. The language used also needs to be revised in order to follow an academic style of writing. 

Author Response

Response to Reviewer 3 Comments

We feel great thanks for your professional review work on our article. As you are concerned, there are several problems that need to be addressed. According to your nice suggestions, we have made extensive corrections to our previous manuscript. The detailed point-by-point responses are listed below.

Point 1: The manuscript introduces to a large extent the principles behind the framework that is presented but fails to discuss the specific details of the model and how it was developed, making the results not reproducible.

Response 1: Thank you for your valuable feedback and sorry for the inadequate description here. We appreciate your comments and acknowledge the importance of providing specific details of the model and how it was developed to ensure the reproducibility of the results. In response to your suggestion, we will revise the manuscript to provide more specific details about the model development process, which will help readers to better understand our approach and reproduce our results. We have made the following modifications and additions aboat the specific details of model [Pg7, Line250-274]:

" The digital twin framework of CNC systems based on the hybrid model mainly consists of two parts: the physical space and the cyber space. The physical space and cyber space interact and integrate virtual and real data through digital threads (such as OPC UA and MTConnect) and the IoT technologies, based on different task re-quirements. The specific details are described as follows:

Physical space: It primarily includes the physical entities of the control system, electromechanical system, and mechanical system. In this space, the control system sends control signals to the electromechanical system, receives feedback from it, and the electromechanical system outputs torque information to the mechanical system while responding to changes in load such as working load and moment of inertia.

Cyber space: This space mainly comprises five components: kinematic chain model, dynamic model, digital twin database, AI algorithms and virtual machine tool. The kinematic chain model and dynamic model represent the mechanical system and electromechanical system virtually through a multidisciplinary unified modeling ap-proach. Detailed descriptions of these models will be provided in sections 3.3 and 3.4. Some typical intelligent functions of CNC systems, such as machining trajectory error compensation and prediction, virtual debugging, and fault diagnosis, require data analysis and storage. Therefore, a digital twin database is used to store information generated during the machining process in both the physical and virtual spaces, and AI algorithms are utilized for data analysis. The virtual machine tool includes the dig-ital twin model of the machine tool and simulation of the machining process control system. These components simulate the actual machining process in the cyber space, and combined with virtual mapping strategies, they optimize and debug machining process parameters in the physical space based on simulation results, achieving per-formance optimization in complex machining scenarios of CNC systems. "

We add the following to the description of the development process of the model [[Pg8, Line275-302]]:

"As a typical mechatronics product, the CNC system involves various disciplines such as mechanical engineering, electrical engineering, and control system engineering. It exhibits characteristics such as multivariable, nonlinearity, and strong coupling, making the modeling and simulation of CNC systems particularly challenging. Tradi-tional single-domain simulation tools are insufficient to meet the requirements for an-alyzing the overall performance of complex systems. Therefore, it is necessary to em-ploy multi-domain modeling and simulation technology to complete the model devel-opment process. Additionally, we enhance the analysis and decision-making capabili-ties of the models through digital twin technology and artificial intelligence. The steps for model development are as follows:

1.Decomposition of the overall system: Considering the relationships among vari-ous components of the CNC system in the physical space, the system is decom-posed into three subsystem models: mechanical, electrical, and control. Mechanis-tic analysis is performed on these subsystems.

2.Construction of the information space: Based on the mechanisms of the subsys-tems in the physical space, the role of the information space in the intelligent functions of the CNC system is analyzed. Using the Modelica multi-domain uni-fied modeling language, the operational mechanisms of each subsystem are com-piled and described to establish a digital twin database. This model is then up-dated and optimized through AI algorithms and machining simulation of the virtual machine tool, achieving the virtual-to-real mapping between the physical and information spaces. This process results in the creation of the digital twin model of the CNC system, ensuring good consistency between the physical opera-tion and model response.

3.Communication between subsystems: The coupling relationships among the sub-systems are analyzed, and their coupling mechanisms are studied to construct coupling interfaces between the subsystems. Digital threads, IoT, and other tech-nologies are utilized to realize the coupling connections between the subsystems, thereby establishing the digital twin framework of the CNC system based on the hybrid model."

Thank you again for your insightful feedback.

Point 2: Details on the toolpaths tested, the result of the machining process and the measured deviations from the nominal profile programmed are entirely missing.

Response 2: Thank you for the reminder. According to your suggestion, we have supplemented the information about the toolpaths tested, the result of the machining process, and the measured deviations from the nominal profile programmed using a combination of images, tables, and descriptive text. We make the following supplement about the toolpaths tested and the result of the machining process [Pg24, Line809-817]:

" The experimental results of the machining trajectory path including the command po-sition of the CNC system and the feedback position of the servo drive are shown in Figure 18. At the same time, in order to ensure the robustness of the introduced method, this paper divides the collected machining trajectory into two parts for verification. Trajectory 1 mainly contains the tool path of the arc, and trajectory 2 is the tool path of the combination of line and arc with corner. "

" Figure 18. Theoretical machining trajectory and actual machining trajectory in the experiments."

We make the following supplement about the the measured deviations from the nominal profile programmed [Pg28, Line889-898]:

" In trajectory 1, the maximum value is 0.169 mm, and the mean value is 0.096 mm before the compensation using the method. The maximum value of the error is reduced to 0.082 mm, and the mean value is reduced to 0.041 mm after the compensation. The maximum value of the error of the machining trajectory is reduced by 51.47%, and the mean value is reduced by 57.29%. In trajectory 2, the maximum value is 0.173 mm, and the mean value is 0.081 mm before the compensation using the method. The maximum value of the error is reduced to 0.083 mm, and the mean value is reduced to 0.033 mm after the compensation. The maximum value of the error of the machining trajectory is reduced by 52.02%, and the mean value is reduced by 59.26%. "

Point 3: The findings of the model need to be verified with multiple toolpaths to ensure the robustness of the approach presented.

Response 3: Thank you for the precious comment. We acknowledge the importance of verifying the findings of the model with multiple toolpaths to ensure the robustness of the approach presented. We have conducted extensive verification by implementing the model across a diverse set of toolpaths. This process involved the meticulous analysis of the model's performance and behavior under varying toolpath configurations. We complements the experiments by using tool paths that contain only circular arcs (trajectory 1) and tool paths that include combinations of straight lines and circular arcs (trajectory 2). The method is improved by pictures and experimental data. The modifications and additions are as follows[Pg25, Line842-845, Pg26, Line851-853, Pg25, Line856-860, Pg27, Line866-868]:

"Figure 19. (a) The loss function of X-axis tracking error prediction results for trajectory 1; (b) The loss function of Z-axis tracking error prediction results for trajectory 1; (c) The loss function of X-axis tracking error prediction results for trajectory 2; (d) The loss function of X-axis tracking error prediction results for trajectory 2. "

"Figure 20. (a) Predicted and true values of X-axis tracking error for trajectory 1; (b) Predicted and true values of Z-axis tracking error for trajectory 1; (c) Predicted and true values of X-axis tracking error for trajectory 2; (d) Predicted and true values of Z-axis tracking error for trajectory 2. "

" The maximum value of the X-axis tracking error prediction is 77.94 μm in trajectory 1. The maximum value of the Z-axis tracking error prediction is 0.34 μm in trajectory 1. The maximum value of the X-axis tracking error prediction is 0.96 μm in trajectory 2. The maximum value of the Z-axis tracking error prediction is 0.0076 μm in trajectory 2.  "

"Figure 21. (a) Prediction error of X-axis tracking error for trajectory 1; (b) Prediction error of Z-axis tracking error for trajectory 1; (c) Prediction error of X-axis tracking error for trajectory 2; (d) Prediction error of Z-axis tracking error for trajectory 2. "

"Figure 21. (a) Comparison of uncompensated and compensated errors of machining trajectory 1; (b) Comparison of uncompensated and compensated errors of machining trajectory 2. "

Reviewer 4 Report

Comments and Suggestions for Authors

This work proposed a compensation and error reduction method in machinery employing digital twins with artificial intelligence algorithms applied to computer numerical control (CNC). The authors tested the proposed method used to a digital twin.  

The paper overall is well-written and organized. The recommendations are: 

1. Lines 56-62, “The advantage of this method is that it can more accurately describe the change rule of machining trajectory error. However, the factors of model accuracy have a significant impact on the prediction and compensation results, especially since the mathematical model cannot accurately reflect the mapping relationship between the machining trajectory error and a variety of influencing factors such as motion control parameters, contour shape, CNC machining performance, etc. Establishing the model and optimizing the parameters requires a higher degree of sophistication.” Please provide references for this statement. 

2. Lines 63-71, Please also provide references to these statements.

3. Lines 100 and 101. Remove “First and Second bullet”.

4.  Add a reference for Figure 1. 

5. Why does the author use the “GB/T 40647-2021 "Intelligent 126 Manufacturing System Architecture" besides other well-known reference models such as the Reference Architecture Model for Industry 4.0 (RAMI 4.0) or Industrial Internet Reference Architecture (IIRA) model?

6.  Line 275, “Fig. 5.” line 322, “Fig. 6.”, line 368, “Fig. 8.”, line 534, “Fig. 10.”, line 727, “Fig. 15.”, line 752, “Fig. 17.”, line 777, “Fig. 18.”, line 788, “Fig. 20.”, and line 807, “Fig. 21.”        

7.  I suggest avoiding the use of “etc”. If the reader is not from the field can miss important information. Also, for me, as a reader, it means that the authors are lazy in writing the information. For example, lines 475 and 476.

8.   Lines 540-541, “Compared to traditional recurrent neural networks, the TTTEP model can better handle long-distance dependencies and avoid gradient vanishing and exploding problems in the recursive structure. As a result, it performs better in data prediction tasks.” Please add a reference to prove this statement. 

9.     Same as item 7 to lines 549-550.

10.  Since the authors used OPC UA architecture and Wireshark to study communication, it is possible to extract the communication latency. What was the latency from the digital twin to the physical device and vice-versa? 

Comments on the Quality of English Language

Minor English revision.

Author Response

Response to Reviewer 4 Comments

We feel great thanks for your professional review work on our article. As you are concerned, there are several problems that need to be addressed. According to your nice suggestions, we have made extensive corrections to our previous manuscript. The detailed point-by-point responses are listed below.

Point 1: Lines 56-62, “The advantage of this method is that it can more accurately describe the change rule of machining trajectory error. However, the factors of model accuracy have a significant impact on the prediction and compensation results, especially since the mathematical model cannot accurately reflect the mapping relationship between the machining trajectory error and a variety of influencing factors such as motion control parameters, contour shape, CNC machining performance, etc. Establishing the model and optimizing the parameters requires a higher degree of sophistication.” Please provide references for this statement.

Response 1: Thank you for your valuable suggestion. We added references to this part of the explanation. We revised the sentence as follows [Pg2, Line60]:

"The advantage of this method is that it can more accurately describe the change rule of machining trajectory error. However, the factors of model accuracy have a significant impact on the prediction and compensation results, especially since the mathematical model cannot accurately reflect the mapping relationship between the machining tra-jectory error and a variety of influencing factors such as motion control parameters, contour shape, CNC machining performance, etc. Establishing the model and optimizing the parameters requires a higher degree of sophistication [13]."

Point 2: Lines 63-71, Please also provide references to these statements.

Response 2: Thank you for your valuable suggestion. We added references to this part of the explanation. We revised the sentence as follows [Pg2, Line70]:

"Data-based methods, on the other hand, utilize data analysis techniques for error prediction and compensation by collecting data from the actual machining process, such as machining process data, cutting force, etc. Standard methods include regression analysis, neural networks, and so on. The advantage of this method is that it can make full use of the information of the actual data and has better adaptability to com-plex error characteristics. Still, on the one hand, it requires a large amount of experi-mental data and robust data processing capabilities. On the other hand, it is affected by the sampling frequency of the data. The problems of inaccuracy of the prediction and compensation will occur in the low sampling frequency [14]."

Point 3: Lines 100 and 101. Remove “First and Second bullet”.

Response 3: Thank you for your valuable suggestion. We removed “First and Second bullet”. We revised the sentence as follows[Pg3, Line99-101]:

" A hybrid-model-based digital twin framework is proposed for CNC systems

A neural-network-model-based machining trajectory tracking error prediction algorithm is proposed."

Point 4: Add a reference for Figure 1

Response 4: Thank you for your valuable suggestion. We added a reference to Figure 1[Pg3, Line131].

"Figure 1. The changes of CNC systems in intelligent manufacturing [24]. "

Point 5: Why does the author use the “GB/T 40647-2021 "Intelligent 126 Manufacturing System Architecture" besides other well-known reference models such as the Reference Architecture Model for Industry 4.0 (RAMI 4.0) or Industrial Internet Reference Architecture (IIRA) model?

Response 5: Thank you very much for the question. After reviewing well-known reference models such as the Reference Architecture Model for Industry 4.0 (RAMI 4.0) or Industrial Internet Reference Architecture (IIRA) model, we have chosen to use the GB/T 40647-2021 "Intelligent Manufacturing System Architecture" as the reference model in this paper for the following reasons:

  1. In the field of intelligent manufacturing research, with the passage of time and technological innovations, the architecture related to intelligent manufacturing is continuously evolving. Compared to the previous two models, the "GB/T 40647-2021 Intelligent Manufacturing System Architecture" referenced in this paper is more recent, thus attracting more attention and discussion. It is also more cutting-edge and practical, which helps in building a comprehensive and coherent research framework.
  2. In contrast to other models, the intelligent manufacturing system architecture referenced in this paper describes the elements, equipment, activities, etc., related to intelligent manufacturing from three dimensions: lifecycle, system levels, and intelligent features. It provides a clearer explanation and description of the standardized objects and scope of intelligent manufacturing.
  3. Furthermore, the referenced standard in this paper is the latest national standard on intelligent manufacturing system architecture issued by the Standardization Administration of China. By referencing this standard, we aim to ensure the accuracy of the framework proposed in this paper to better meet the requirements and standards in the field of intelligent manufacturing.

These reasons collectively justify our choice to utilize the "GB/T 40647-2021 Intelligent Manufacturing System Architecture" as the primary reference model in this study.

Point 6: Line 275, “Fig. 5.” line 322, “Fig. 6.”, line 368, “Fig. 8.”, line 534, “Fig. 10.”, line 727, “Fig. 15.”, line 752, “Fig. 17.”, line 777, “Fig. 18.”, line 788, “Fig. 20.”, and line 807, “Fig. 21.”

Response 6: Thank you especially for your suggestion, which is due to a formatting problem caused by our oversight. After correcting these errors that you raised, we checked to make sure that these errors do not occur in this article.

Point 7: I suggest avoiding the use of “etc”. If the reader is not from the field can miss important information. Also, for me, as a reader, it means that the authors are lazy in writing the information. For example, lines 475 and 476.

Response 7: Thank you for your valuable suggestion. We made changes to the article about "etc" to ensure that no information is implied or assumed and that the content is accurate. We revised the sentence as follows [Pg15, Line500-502]:

" The machining trajectory data contains the command position, command speed and command acceleration, generated by the CNC system, and the actual position, actual speed, actual acceleration and tracking error, fed back by the servo drive."

 Point 8: Lines 540-541, “Compared to traditional recurrent neural networks, the TTTEP model can better handle long-distance dependencies and avoid gradient vanishing and exploding problems in the recursive structure. As a result, it performs better in data prediction tasks.” Please add a reference to prove this statement.

Response 8: Thank you for your valuable suggestion. We added references to this part of the explanation. We revised the sentence as follows [Pg17, Line569]:

"Compared to traditional recurrent neural networks, the TTTEP model can better handle long-distance dependencies and avoid gradient vanishing and exploding problems in the recursive structure. As a result, it performs better in data prediction tasks [36]."

Point 9: Same as item 7 to lines 549-550.

Response 9: Thank you for your valuable suggestion. We revised the sentence as follows [Pg15, Line520-522]:

" Traditional feature extraction mainly includes the position information in respect of each axis, curvature, speed, acceleration."

Point 10: Since the authors used OPC UA architecture and Wireshark to study communication, it is possible to extract the communication latency. What was the latency from the digital twin to the physical device and vice-versa?

Response 10: Thank you very much for the question. As you mentioned, the communication delay between devices is crucial in the experiment. The Beckhoff network analyzer used in this study is the ET2000 model, which provides high-precision timestamping for data frames (bi-directional transmission) with a resolution of 1 ns, enabling extremely accurate time analysis of the connected network segments. According to the product specifications of the device (https://www.beckhoff.com.cn/zh-cn/products/i-o/ethercat-development-products/elxxxx-etxxxx-fbxxxx-hardware/et2000.html), its low cycle delay of 1 µs results in minimal impact on the system.

In the experimental process, we observed that the communication delay between the CNC and PC during data acquisition is very small (≤2ms), and therefore can be considered negligible. Furthermore, the data acquisition method employed in this study is based on the actual machining process of the machine tool. The error prediction and compensation model are trained offline, and the latency between the Beckhoff network analyzer and the PC is not a significant concern when using this offline model during the machining process.

Round 2

Reviewer 2 Report

Comments and Suggestions for Authors

The authors have adequately addressed the observations made in the previous version of the document. I consider that the work can be published in its current form.

Comments on the Quality of English Language

The authors have adequately addressed the observations made in the previous version of the document. I consider that the work can be published in its current form.